

# Satellite Passive Microwave Sea-Ice Concentration Data Set Inter-comparison using Landsat data

Stefan Kern[1], Thomas Lavergne[2], Leif Toudal Pedersen[3], and Rasmus Tage Tonboe[4a], Louisa Bell[1b], Maybritt Meyer[1c], and Luise Zeigermann[1d]

[1]Integrated Climate Data Center (ICDC), Center for Earth System Research and Sustainability (CEN), University of Hamburg, Hamburg, Germany
[2]Research and Development Department, Norwegian Meteorological Institute, Oslo, Norway
[3]Danish Technical University, Lyngby, Denmark
[4]Danish Meteorological Institute, Copenhagen, Denmark
[a]now at Danish Technical University, Lyngby, Denmark
[b]now at Climate Service Center Germany (GERICS), Helmholtz-Zentrum Hereon, Hamburg, Germany
[c]now at Federal Maritime and Hydrographic Agency, Hamburg, Germany
[d]now at Faculty of Science for Physics and Physical Oceanography, Memorial University of Newfoundland, St. Johns, Canada

*Correspondence to*: Stefan Kern (stefan.kern@uni-hamburg.de)

**Abstract.** We report on results of an inter-comparison of 10 global sea-ice concentration (SIC) data products at 12.5 to 50.0 km grid resolution from satellite passive microwave (PMW) observations. For this we use SIC estimated from > 300 images acquired in the visible / near-infrared frequency range by joint the National Aeronautics and Space Administration (NASA)/United States Geological Survey (USGS) Landsat sensor during years 2003-2011 and 2013-2015. Conditions covered are late winter / early spring in the Northern Hemisphere and from late winter through fall freeze-up in the Southern Hemisphere. Among the products investigated are the four products of the European Organisation for the Exploitation of Meteorological Satellites (EUMETSAT) Ocean and Sea Ice Satellite Application Facility (OSI SAF) and European Space Agency (ESA) Climate Change Initiative (CCI) algorithms: SICCI-2 and OSI-450. We stress the importance to consider inter-comparison results across the entire SIC range instead of focusing on overall mean differences, and to take in account known biases in PMW SIC products, e.g. for thin ice. We find superior linear agreement between PMW SIC and Landsat SIC for the 25 km and the 50 km SICCI-2 products in both hemispheres. We discuss quantitatively various uncertainty sources of the evaluation carried out. First, depending on the number of mixed ocean-ice Landsat pixels classified erroneously as ice only, our Landsat SIC is found to be biased high. This applies to some of our Southern Hemisphere data, promotes an overly large fraction of Landsat SIC under-estimation by PMW SIC products, and renders PMW SIC products overestimating Landsat SIC particularly problematic. Secondly, our main results are based on SIC data truncated to the range 0 % to 100 %. We demonstrate using non-truncated SIC values, where possible, can considerably improve linear agreement between PMW and Landsat SIC. Thirdly, we investigate the impact of filters often used to clean up the final products from spurious SIC over open water due to weather effects and along coastlines due to land spillover. Benefiting from the possibility to switch on or off certain filters in the SICCI-2 and OSI-450 products we quantify the impact land spillover filtering can have on evaluation results as shown in this paper.

## 1 Introduction

We carry on the evaluation of sea-ice concentration (SIC) products derived from satellite passive microwave (PMW) observations. In Kern et al. (2019), we presented an evaluation of ten PMW SIC products at 0 % and 100 % SIC, and with respect to sea-ice observations along ship tracks. Another study focused on Arctic summer conditions, investigating the bias between these PMW SIC products and independent SIC and net ice surface fraction estimates based on MODerate resolution Imaging Spectroradiometer (MODIS) observations (Kern et al., 2020). With this study, we shift our focus more towards intermediate SIC and utilize are much larger and, partly, more accurate reference dataset than in the two earlier studies. The





evaluation at 0 % SIC in Kern et al. (2019) used a few fixed open water locations only. The evaluation at 100 % SIC used
near-100 % SIC estimates based on the analysis of freezing-season synthetic aperture radar (SAR) image pairs of convergent
high-concentration ice situations. With that we evaluated the PMW SIC products for one specific set of ice conditions only
(winter and near 100 %). Kern et al. (2019) also presented results of an evaluation of PMW SIC against a multi-annual set of
standardized manual visual ship-based observations of the ice conditions. These observations are, however, of limited accuracy
and of limited representativity because the average accuracy is between 5 % and 10 % and observations mostly represent sea-
ice conditions where it is possible to navigate. In addition, to reduce noise, PMW and ship-based SIC were averaged over all
observations along a ship-track within one day, representing sea-ice conditions across spatial scales, that – in the worst case –
vary by an order of magnitude. The averaging resulted in a reduction of the number of valid data pairs from about 15000 to
less than 800, i.e. about 400 per hemisphere.

Another aspect is that the accuracy of the hemispheric but also the regional sea-ice area (SIA) computed from PMW

SIC estimates strongly depends on their accuracy. PMW SIC values biased high yield an overestimation of the SIA whereas
PMW SIC biased low results in an underestimation of the SIA. This seems not to be critical as long as the trend is correct (e.g.
Ivanova et al., 2014) but limits the use of such SIA estimates for quantitative inter-comparisons of climate-model results
against observations (e.g. Burgard et al., 2020). It is for sure important PMW SIC is 100 % where the actual SIC is 100 % to
avoid artificially elevated ocean-atmosphere heat flux when used as a surface forcing. It is equally important PMW SIC is an
accurate estimate of the open water fraction, hence providing 95 % where the actual SIC is 95 % due to leads and openings in
the sea-ice cover. In addition, it is desirable to check the performance of PMW SIC products across the entire SIC range in
order to have a reliable estimate of the actual ice cover in, for example, the marginal ice zone (MIZ). Here gradients in heat
fluxes are particularly pronounced and small changes in the SIC can have a comparably large impact on ocean-atmosphere
heat transfer. A correct definition of and accurate SIC distribution within the MIZ are also crucial should SIC values be used
to evaluate numerical models capable to simulate wave-sea ice interaction (e.g. Boutin et al., 2020; Nose et al., 2020). The
ship-based SIC observations used in Kern et al. (2019) offer only limited potential to carry out this performance check because
of i) their accuracy and limitations in spatial representativity, ii) small number of observations falling into the relevant SIC
range of, e.g. 20 % to 80 %, and iii) the larger observational error in this SIC range.

Therefore, in this paper we focus on the evaluation of PMW SIC products against a large number of high-resolution

binary sea-ice cover maps estimated from satellite observations acquired in the visible frequency range by NASA/USGS
Landsat-5, 7 and 8. Overall, we used over 300 such Landsat-based maps, corresponding to more than 10 000 25 km x 25 km
resolution PMW SIC grid cells. We chose Landsat over MODIS because of the substantially finer spatial resolution of the
visible channels of Landsat: 30 m compared to MODIS: 250 m. Another option would have been to use Sentinel-2
MultiSpectral Instrument (MSI) (Drusch et al., 2012). We discarded this option in light of the limited overlap between this
satellite mission (Sentinel-2A was launched June 2015) and our PMW SIC data set but it will be very valuable in the future
since it will allow extending the dataset to areas much further from land and will likely provide an even more accurate
evaluation data set.

Utilization of the high-resolution information provided by the Landsat satellites as a means for assessing satellite

PMW SIC products dates back to the early 1980ties when Comiso and Zwally (1982) compared Nimbus-7 Scanning
Multichannel Microwave Radiometer (SMMR) SIC with Landsat imagery. Since then a number of studies used a small number
of such images for inter-comparison and/or evaluation studies of SIC retrievals (e.g. Steffen and Maslanik, 1988; Steffen and
Schweiger, 1991; Comiso and Steffen, 2001; Cavalieri et al., 2006; Wiebe et al., 2009; Lu et al., 2018; Zhao et al., 2021).
Landsat imagery has also recently been used for quality assessment of SIC estimates from Suomi/NPP VIIRS observations
(e.g. Liu et al., 2016). Common to all these studies is they used a comparably small number of Landsat scenes, i.e. less than
ten, an order of magnitude smaller than the number of scenes used in this study (see above).





Analysis of visible satellite imagery for SIC estimation is quite straightforward. A threshold based method
discriminating between open water and ice is applied at the native spatial resolution (pixel size: 30 m x 30 m) of the Landsat
channels in the visible frequency range, assuming that a pixel is covered by either ice or water. Co-locating this high-resolution
information of the binary ice-water distribution with the coarse-resolution PMW SIC products and counting ice and water
pixels within a PMW SIC product's grid cell provides an adequate independent measure of the SIC. We refer to Section 2.2
for more details.
For evaluating the PMW SIC products across the SIC range, we prefer to use visible data instead of SAR data. The
main advantages of SAR data would be i) the larger area covered by a single scene compared to Landsat (about 400 km to 500
km in SAR wide-swath mode (WSM) vs. 180 km for Landsat) and ii) their independence to daylight and cloud cover. In fact,
many PMW SIC inter-comparison studies have already used SAR images (e.g., Comiso et al., 1991; Dokken et al., 2000;
Belchansky and Douglas, 2002; Kwok, 2002; Heinrichs et al., 2006; Andersen et al., 2007; Wiebe et al., 2009; Han and Kim,
2018). However, despite the past decade's substantial progress in developing and testing methods to translate SAR images into
high-resolution SIC maps (e.g.: Cooke and Scott, 2019; Karvonen, 2014, 2017; Komarov and Buehner, 2017, 2019; Leigh et
al., 2014; Lohse et al., 2019; Ochilov and Clausi, 2012; Singha et al., 2018; Wang et al., 2016, 2017; Zakhvatkina et al., 2017,
Boulze et al., 2020; Malmgren-Hansen et al., 2020; Wang and Li, 2020), some using machine learning approaches, the accuracy
of the obtained SIC maps is not always satisfying. Particularly at intermediate SIC – the main focus of this study – SAR
signatures are often ambiguous, resulting in SAR SIC uncertainties too large for our purposes. Furthermore, applications of
such methods to derive Southern Ocean SIC from SAR are comparably sparse. Therefore, we do not use SAR-based SIC maps.
We note that also Ice charting services (FMI, DMI, MET Norway, CIS, NATICE, AARI) heavily depend on SAR
imagery for production of their ice charts. They thus have a large demand to automate processes of classification and are
potentially most advanced in testing automated SAR SIC retrieval (e.g. Cheng et al., 2020). However, ice charts provide SIC
ranges within polygons highly variable and heterogeneous in size and shape. Several studies used such ice charts for various
inter-comparison purposes (e.g. Shokr and Markus, 2006; Shokr and Agnew, 2013, Titchner and Rayner, 2014). Some centers
providing operational sea-ice information also use such charts for routine quality checking of PMW SIC products. However,
for our purpose evaluating PMW SIC CDRs and similar SIC products, the limitations of such charts in terms of precision and
accuracy – particularly in the intermediate SIC range (e.g. Cheng et al., 2020), exclude their usage in this study.
After this introduction, this paper provides information about the PMW SIC products, the Landsat data set used and
the methods applied to derive SIC from the Landsat images (Sect. 2). We present our results in Sections 3 and 4, discuss some
additional aspects in Section 5 and conclude the study in Section 6.
**2    Data & Methodologies**
**2.1    Sea-ice concentration data sets**
The ten different PMW SIC products considered in our study are summarized briefly in Table 1. We refrain from
repeating information about the algorithms themselves, tie point selection, application of weather filters, consideration of land
spillover effects and so forth. All this information is provided in detail in Lavergne et al. (2019), Kern et al. (2019, Appendix
7.1-7.6), and Kern et al., (2020). The same applies to the fact that four of the products (SICCI-12km, SICCI-25km, SICCI-
50km, and OSI-450) allow to take into account the full SIC distribution at the two end-member sea-ice concentrations: 0 %
and 100 % which naturally result from the SIC retrieval method used in all considered SIC products but the NT2-AMSR
product. This distribution contains negative as well as above-100 % SIC values that are typically truncated, i.e. set to the
exactly 0 % and 100 %. We refer to Lavergne et al. (2019) and Kern et al. (2019) for more information in this regard.
In order to extend the time-series of the Comiso Bootstrap (CBT) algorithm and the NASA-Team 2 (NT2) algorithm
using Advanced Microwave Scanning Radiometer aboard Earth Observation Satellite (AMSR-E) data beyond its lifetime



(2011-10-03), we use the respective unified product based on data of the Advanced Microwave Scanning Radiometer aboard
GCOM-W1: AMSR2 and of AMSR-E (Meier et al., 2018). With that we use five products based on AMSR-E and AMSR2
data and five products based on Special Sensor Microwave / Imager: SSM/I, and Special Sensor Microwave Imager and
Sounder: SSMIS data, of the period 2002 through 2015. We do not use PMW SIC data of the period October 2011 through
July 2012 because of the gap between AMSR-E and AMSR2. All PMW SIC data have daily temporal resolution. The grid
type and grid resolution of all datasets are provided in Table 1.

**Table 1.** Overview of the investigated PMW SIC products. Column "ID (Algorithm)" holds the identifier we use henceforth
to refer to the data product, and which algorithm it uses. Note that for those algorithms where an AMSR sensor forms part of
the name, we refer to AMSR-E or AMSR2, depending on which of the two sensors provides the data. Column "Input data"
refers to the input satellite data for the data set, together with the frequencies and respective field-of-view dimensions.

| ID (algorithm) | Input data; frequencies (field-of-views) | Grid resolution & type | Reference |
|---|---|---|---|
| OSI-450 (SICCI2) | SSM/I, SSMIS; 19.35 GHz (69 km x 43 km), 37.0 GHz (37 km x 28 km) | 25 km x 25 km EASE2.0 | Tonboe et al., 2016; Lavergne et al., 2019 |
| SICCI-12km (SICCI2) | AMSR-E/AMSR2; 18.7 GHz (27 km x 16 km/ 22 km x 14 km), 89.0 GHz (6 km x 4 km/ 5 km x 3 km) | 12.5 km x 12.5 km EASE2.0 | Lavergne et al., 2019 |
| SICCI-25km (SICCI2) | AMSR-E/AMSR2; 18.7 GHz (27 km x 16 km/ 22 km x 14 km), 36.5 GHz (14 km x 8 km/ 12 km x 7 km) | 25 km x 25 km EASE2.0 | Lavergne et al., 2019 |
| SICCI-50km (SICCI2) | AMSR-E/AMSR2 6.9 GHz (75 km x 43 km/ 62 km x 35 km), 36.5 GHz (14 km x 8 km/ 12 km x 7 km) | 50 km x 50 km EASE2.0 | Lavergne et al., 2019 |
| CBT-SSMI (Comiso bootstrap) | SSM/I, SSMIS; 19.35 GHz (69 km x 43 km), 37.0 GHz (37 km x 28 km) | 25 km x 25 km PolarStereo | Comiso, 1986; Comiso et al., 1997; Comiso and Nishio, 2008 |
| NOAA-CDR (NASA Team & Comiso bootstrap) | SSM/I, SSMIS;19.35 GHz (69 km x 43 km), 37.0 GHz (37 km x 28 km) | 25 km x 25 km PolarStereo | Peng et al., 2013; Meier and Windnagel, 2018 |
| CBT-AMSR (Comiso bootstrap) | AMSR-E/AMSR2; 18.7 GHz (27 km x 16 km/ 22 km x 14 km), 36.5 GHz (14 km x 8 km/ 12 km x 7 km) | 25 km x 25 km PolarStereo | Comiso et al., 2003; Comiso and Nishio, 2008; Comiso, 2009 |
| ASI-SSMI (ASI) | SSM/I, SSMIS; 85.5 GHz (15 km x 13 km) | 12.5 km x 12.5 km PolarStereo | Kaleschke et al., 2001; Ezraty et al., 2007 |
| NT1-SSMI (NASA-Team) | SSM/I, SSMIS;19.35 GHz (69 km x 43 km), 37.0 GHz (37 km x 28 km) | 25 km x 25 km PolarStereo | Cavalieri et al, 1984; 1992; 1999 |
| NT2-AMSR (NASA-Team-2) | AMSR-E/AMSR2; 18.7 GHz (27 km x 16 km/ 22 km x 14 km), 36.5 GHz (14 km x 8 km/ 12 km x 7 km), 89.0 GHz (6 km x 4 km/ 5 km x 3 km) | 25 km x 25 km PolarStereo | Markus and Cavalieri, 2000; 2009 |


**2.2    The Landsat data set**
Landsat data of the Thematic Mapper TM on Landsat-5, the Enhanced Thematic Mapper (ETM) on Landsat-7, and
Operational Land Imager (OLI) on Landsat-8 were obtained in Level 1c GeoTIFF format from https://earthexplorer.usgs.gov
[last accessed: June 28, 2021] for years 2003-2011 (Landsat-5), 2003 (Landsat-7), and 2013-2015 (Landsat-8). Only images
with a cloud fraction < 30 % provided as a search criterion upfront, were selected and downloaded from the server. In the
Northern Hemisphere, we use images of months March, April, May and September, i.e. from late winter to spring and at the
onset of fall freeze-up; in the Southern Hemisphere we use images of months October through March, i.e. from late winter
over summer to fall freeze-up. The total number of images acquired is 421; these split into 152, 12, and 227 for Landsat-5, 7
and 8, respectively, and partition into 259 images for the Northern Hemisphere and 162 images for the Southern Hemisphere.





### 2.2.1 Processing
We compute the top of atmosphere (TOA) reflectance for channels 2 to 4 (Landsat-5 and 7) or channels 3 to 5
(Landsat-8) following Chander et al. (2007; 2009) and USGS (2015). Table 2 provides the wavelengths of the channels used
(e.g. Chander et al., 2009; Barsi et al., 2014). The solar zenith angle and other parameters required for this computation is
either included in the Landsat data files or is taken from Chander et al. (2007, 2009) and the Landsat 8 data users handbook
(USGS, 2015). To convert the TOA reflectances to surface reflectances or surface albedo we follow the approaches of Koepke
(1999) and Knap et al. (1999) that assume that the TOA reflectance (or planetary reflectance) equals the TOA albedo (or
planetary albedo) and that the TOA albedo $\alpha_{TOA}$ is related to the surface albedo $\alpha_{surface}$ via the simple linear relationship:
$$\alpha_{TOA} = a + b\alpha_{surface} \quad (1)$$
The coefficients $a$ and $b$ are a function of the atmospheric conditions, the solar zenith angle, and the wavelength. We follow
Koepke (1999) and take values for $a$ and $b$ from his figure 1 (KF1) and figure 2 (KF2). KF1 derived for the Advanced Very
High Resolution Radiometer (AVHRR) channel 1 we use for Landsat channels in the wavelength range 500-700 nm. KF2
derived for AVHRR channel 2 we use for Landsat channels in the wavelength range 700-900 nm. We choose those atmospheric
conditions that are appropriate for a polar marine atmosphere. For aerosol optical depth we use 0.05, for ozone content we use
0.24 cm[NTP] (NTP stands for normal temperature and pressure) corresponding to 240 Dobson Units, and for water vapor
content we used 0.5 g/cm². Using Eq. (1) we convert TOA albedo into surface albedo values separately for the three channels
of the respective Landsat instrument. Subsequently, we compute from these surface albedo values an estimate of the surface
broadband shortwave albedo (e.g. Brandt et al., 2005) using the bandwidths of the channels as weights. The change in
bandwidths between the Landsat instruments is thus taken into account (see Table 2).

**Table 2.** Overview about the wavelengths of the Landsat channels used.

| Wavelength [nm] of | Landsat-5 | Landsat-7 | Landsat-8 |
|---|---|---|---|
| Channel 2 | 528-609 | 519-601 | -- |
| Channel 3 | 626-693 | 631-692 | 533-590 |
| Channel 4 | 776-904 | 772-898 | 636-673 |
| Channel 5 | -- | -- | 851-879 |


For every broadband surface albedo map, we perform a supervised visual classification into open water, bare / thin
ice and snow covered / thick ice. For that, we assume the respective surface class covers a Landsat pixel entirely. We assign
all dark pixels (with an albedo of, on average, smaller than 0.06) to the open water class. We assign all bright pixels (with an
albedo of, on average, larger than 0.45) to the class snow covered / thick ice; all remaining pixels fall into the class bare / thin
ice. We pay more attention separating open water from ice very accurately than to distinguish between bare / thin ice and
snow-covered / thick ice. In every Landsat albedo map we search for leads or openings, zoom into these and perform histogram-
equalized slicing to visually identify – based on albedo values and spatial structures – whether the leads or openings selected
contain open water. The threshold value chosen to separate open water from ice we take from Pegau and Paulsen (2001). The
threshold value chosen to distinguish between bare / thin ice and snow covered / thick ice is based on Brandt et al. (2005) and
Zatko and Warren (2015). They found an albedo of around 0.33 for bare thin ice less than 30 cm thick and of around 0.42 for
snow covered thin ice (5 - 10 cm thick) with a thin (< 3 cm) snow cover. Note that the actual threshold values chosen for a
particular Landsat image varies between 0.03 and 0.08 for the open water – ice discrimination and between 0.35 and 0.55 for
the bare / thin ice – snow covered / thick ice discrimination. This variation results from the varying illumination conditions
encountered – despite our limitation to Landsat scenes acquired at solar zenith angles < 65°.
Usage of a three-class distribution is motivated by the fact that it has been shown that PMW SIC is often biased low
over thin sea ice (e.g. Wensnahan et al., 1993; Cavalieri, 1994; Ivanova et al., 2015). Therefore, in addition to using the Landsat
images just for a high-resolution ice-water discrimination we also use them to derive the fraction of thin ice with the aim to
discuss differences between Landsat SIC and PMW SIC in the light of a potential impact by thin ice. However, we discarded





this aim – but kept the classification results – because during analyses of the Landsat images we encountered ambiguities in
surface albedos between snow-covered thin ice and bare thick ice. While there is little ambiguity between open water and ice,
except for very thin dark nilas or ice rind (e.g. Zatko and Warren, 2015), resulting in high confidence of pixels classified as
either open water or ice, the confidence of pixels classified as bare/thin or snow covered/thick ice is considerably worse.

***2.2.2 Co-location and comparison***
For the co-location, we first select a rectangular area within the PMW SIC grid, EASE-2 for the SICCI-2 and OSI-
450 products and polar-stereographic true at 70 degrees northern or southern latitude (known as NSIDC grid) for the other six
products, which encloses the Landsat SIC map. For this we take the geographic corner coordinates of the Landsat SIC map
(still at 30 m grid resolution), convert these into Cartesian Coordinates and find those PMW SIC grid cells which centers have
minimum distance (in meters) to these corner coordinates. Beforehand, we also convert PMW SIC grid cell coordinates into
Cartesian coordinates and rotate the grid for the Northern Hemisphere PMW SIC products on the NSIDC grid clockwise by
45 degrees; this is not required for the respective Southern Hemisphere PMW SIC products.

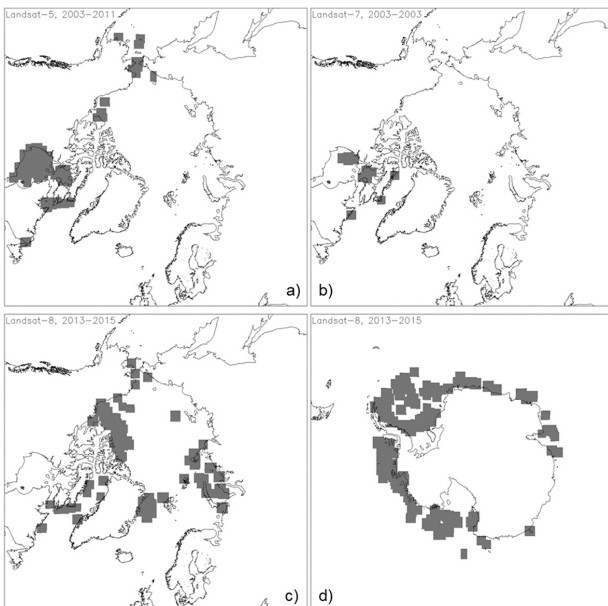

**Figure 1.** Location of the Landsat scenes used. Panels a) through c) Arctic; panel d) Antarctic. Note that scenes do overlap.
The total number of scenes shown is 134 (a), 12 (b), 88 (c), and 134 (d).

Subsequently, we compute the Landsat SIC by summing over all 30 m pixels classified as ice that fall into the PMW
SIC grid cells within the above-defined rectangular area. Because we do this is at the grid resolution of the PMW products, we
obtain Landsat SIC maps at 12.5 km, 25.0 km, and 50.0 km grid resolution. We compare the resulting gridded Landsat SIC
with the respective co-located PMW SIC by computing the mean difference PMW SIC minus Landsat SIC and its standard
deviation, the median difference, and deriving a linear regression line and computing the linear correlation coefficient.
Based on a visual quality check of the obtained Landsat SIC maps we discard quite a number of processed Landsat
scenes from further analysis – mainly because of cloud artifacts but also because a few scenes we obtained twice. Therefore,
the final number of Landsat SIC maps used is lower than indicated above: 234 for the Arctic, partitioning into Landsat-5: 134,

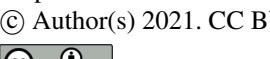



Landsat-7: 12, and Landsat-8: 88, and 134 for the Antarctic. The spatial distribution of the Landsat scenes is illustrated in Fig.
1. Note that we focus on data of Landsat-5 and Landsat-8 in this paper.

*2.2.3 Sensitivity analysis*
In order to estimate how Landsat SIC depends on the choice of the albedo thresholds used to discriminate open water
from ice and bare / thin ice from snow covered / thick ice, we repeat the classification into the three surface classes using
modified thresholds. We vary the albedo value for the open water – ice discrimination by ± 0.03, i.e. for an actual albedo value
of 0.06 we employ additional threshold values of 0.03 and 0.09. We vary the albedo value for the bare / thin ice – snow covered
/ thick ice discrimination by ± 0.1, i.e. for an actual albedo value of 0.45 we employ additional threshold values of 0.35 and
0.55. The range of albedo threshold values we choose is motivated by our experience with the supervised classification of the
many Landsat scenes under varying illumination conditions. We arbitrarily select 12 Landsat 8 scenes for the Northern
Hemisphere, and 15 scenes for the Southern Hemisphere. For every image we perform the classification into the three surface
classes with the above-mentioned four additional albedo threshold value combinations, compute Landsat SIC on the 25 km
and 50 km EASE grid and derive a Landsat scene mean SIC value (Tables 3 and 4). We find that changing the albedo value
of the open water – ice discrimination by ± 0.03 changes the average Landsat SIC by between 0.7 % and 1.2 % in the Northern
Hemisphere and by between 0.8 % and 1.5 % in the Southern Hemisphere. With that the sensitivity appears to be independent
of the overall SIC which is close to 100 % for the Northern Hemisphere cases (Table 3) but 55 – 60 % for the Southern
Hemisphere cases (Table 4). The difference in the sensitivity between grid resolutions of 25 km and 50 km is less than 0.2 %.

**Table 3**. Landsat SIC derived using the actual pair of albedo threshold values ("Actual value") and the four variations of them
(see text) averaged for 12 Landsat-8 scenes selected for the Northern Hemisphere (NH) at 25 km and 50 km grid resolution.
The number behind the ± denotes one standard deviation. All SIC values are in percent.

| $\alpha_{thinice}$ \ $\alpha_{openwater}$ | -0.03 | Actual value | +0.03 | NH, 25km |
|---|---|---|---|---|
| -0.1 | 99.2 ± 2.1 | -- | 97.3 ± 3.7 | |
| Actual value | -- | 98.0 ± 3.1 | -- | |
| +0.1 | 99.2 ± 2.1 | -- | 97.3 ± 3.7 | |
| | | | | NH, 50km |
| -0.1 | 98.9 ± 3.2 | -- | 96.9 ± 4.5 | |
| Actual value | -- | 97.7 ± 4.1 | -- | |
| +0.1 | 98.9 ± 3.2 | -- | 96.9 ± 4.5 | |


**Table 4.** Landsat SIC derived using the actual pair of albedo threshold values ("Actual value") and the four variations of them
(see text) averaged for 15 Landsat-8 scenes selected for the Southern Hemisphere (SH) at 25 km and 50 km grid resolution.
The number behind the ± denotes one standard deviation. All SIC values are in percent.

| $\alpha_{thinice}$ \ $\alpha_{openwater}$ | -0.03 | Actual value | +0.03 | SH, 25km |
|---|---|---|---|---|
| -0.1 | 63.0 ± 27.0 | -- | 60.5 ± 26.4 | |
| Actual value | -- | 61.5 ± 26.6 | -- | |
| +0.1 | 63.0 ± 27.0 | -- | 60.5 ± 26.4 | |
| | | | | SH, 50km |
| -0.1 | 54.5 ± 34.8 | -- | 52.3 ± 33.8 | |
| Actual value | -- | 53.1 ± 34.1 | -- | |
| +0.1 | 54.5 ± 34.8 | -- | 52.3 ± 33.8 | |


As expected, changing the albedo value of the bare / thin ice – snow-covered / thick ice discrimination by ± 0.1 does not
influence the Landsat SIC. However, it influences the Landsat SIC computed at the respective grid resolutions when using
Landsat pixels classified as snow-covered / thick ice only (Tables S02 and S03 in the Supplementary Material). We find
Landsat SIC of thick ice to vary by between 1.4 % and 2.4 % in the Northern Hemisphere and by between 2.1 % and 2.7 % in
the Southern Hemisphere with little difference between the grid resolutions. For the Landsat scenes used in this sensitivity



study in the Northern Hemisphere, we find a difference of 4.8 % between the total SIC and the SIC of pixels classified as
snow-covered / thick ice; hence the average bare / thin ice SIC is 4.8 %. In the Southern Hemisphere, the average bare / thin
ice SIC is 8.8 % at 25 km grid resolution and 7.5 % at 50 km grid resolution (not shown).

***2.2.4 Potential biases in Landsat SIC***
In our approach, we assume either ice or water to cover a Landsat pixel (30 m x 30 m) entirely, not taking into account
that ice floes or leads / openings might be smaller than the pixel size, resulting in a mixed ocean-ice pixel. This can introduce
a positive bias in the Landsat SIC computed at the grid resolution of the PMW SIC products. For instance, for a Landsat pixel
covered just half by snow covered / thick sea ice, which exhibits a surface albedo of 0.8 under cold conditions, the resulting
pixel average albedo would be 0.5 x 0.06 + 0.5 x 0.8 = 0.43. With that, such a pixel is classified as bare / thin ice and counts
as a pixel with 100 % instead of 50 % sea-ice concentration. Depending on the albedo of the ice, an ice-cover fraction of 0.04
in one Landsat pixel could be sufficient to increase the pixel average albedo above the upper open water – ice discrimination
threshold value of 0.09 (see Tables 3, 4), causing the respective pixel to be classified as 100 % ice.
In order to quantify this positive bias better, it is useful to distinguish between sea-ice conditions during summer and
winter, between pack ice and the MIZ, and to take into account the dimensions of leads / openings and ice floes. Distributions
of lead width and floe size both follow a power law. Leads / openings and ice floes with dimensions smaller than the Landsat
pixel size are orders of magnitude more abundant than wide leads / openings (e.g. Tschudi et al., 2002; Marcq and Weiss,
2012) and large ice floes (e.g. Steer et al., 2008; Toyota et al., 2011; Perovich and Jones, 2014).
Based on airborne digital camera visible imagery captured along several thousands' of kilometers long tracks of
Operation Icebridge (OIB) flights in the Arctic in April 2010 and in the Antarctic in October 2009 analyzed by Onana et al.
(2013) with respect to the lead and open water fraction, we find a SIC bias of less than 0.2 %. This value derived for an open
water fraction of ~ 1 % falls into the uncertainty range of our approach (see Tables 3, 4) and represents winter conditions in
the pack ice. Based on manual visual analysis of airborne visible imagery obtained in the MIZ in the Greenland Sea in March
1997, we find a SIC bias of the order of 5 to 10 %. This value is clearly outside the uncertainty range of our approach. The
images used here represent an ice cover of ~ 70 % SIC comprising closely packed but also broken bands of a few thicker ice
floes, pancake ice, brash and grease ice with little or no new ice formation in the openings – a typical situation at an ice edge
located in comparably warm water.
Next, we again take the results of Onana et al. (2013) but assume that the thin ice identified in the OIB digital camera
imagery adds to the open water fraction thereby simulating a summer situation. For an open water fraction of then ~ 5 %, we
estimate a SIC bias of less than 0.8 %, which is still within the uncertainty range of our approach. However, this low positive
bias during summer would only apply to a situation where ice floes are still packed closely together, e.g. by herding of ice
floes (e.g. Toyota et al., 2016), and where gaps between the ice floes from additional openings created by the melt process are
filled by brash ice and/or slush. While this is a situation that might be encountered during summer (Steer et al., 2008; Lu et al.,
2008), it is not necessarily typical. In summer, it can be more common to encounter isolated floes. Depending on the size of
the floes and their distribution across a 25 km grid cell with, e.g., 50 % SIC, we find the bias to range between less than 2 %
to 50 % in the two most extreme cases. We refer to the Supplementary Material to this subsection, where we describe in more
detail how we obtain estimates of the positive bias caused by the combination of i) the finite resolution of the Landsat sensor
and ii) our classification approach for both winter and summer conditions at the scale of a 25 km PMW SIC product grid.
According to the high-resolution optical images used to infer the floe size distribution (Steer et al., 2008; Toyota et
al., 2011; 2016) and similar studies (e.g. Paget et al., 2001; Lu et al., 2008; Zhang and Skjetne, 2015), the ice cover often
comprises a large spectrum of floes. The larger and largest floes at the upper end of the floe-size distribution form the major
fraction of the sea-ice area (in square kilometers) (e.g. Paget et al., 2001; Steer et al., 2008). A small number of large floes
results in a smaller number of mixed ocean-ice Landsat pixels than a large number of smaller floes. Hence, where larger floes



dominate our Landsat SIC estimate is less biased than where small floes dominate. The effect of the ocean swell, the
dominating force for fracturing ice floes according to, e.g., Toyota et al. (2016), is larger close to the ice edge than further
inside the ice pack. Therefore, a larger number of smaller floes exists along the ice edge, suggesting a larger bias in our Landsat
SIC near the ice edge than inside the ice pack. Without further independent information about the actual ice cover, we are not
able quantifying this bias accurately.

In summary, we state: for high-concentration winter conditions and for those cases during summer where ice floes

are closely packed and openings between the floes are covered with brash ice and slush, the bias in Landsat SIC derived at the
spatial scale of the PMW SIC products falls within the retrieval uncertainty range of our approach (see Tables 3, 4). The bias
could fall outside the uncertainty range near the ice edge during winter when sea ice drifts into comparably warm waters that
inhibit ice formation in newly created openings; here biases as high as 10 % in a single PMW grid cell could occur. The bias
could also fall outside the uncertainty range during summer; here biases between 5 % and 20 % in single PMW grid cells might
occur depending on proximity to the ice edge and hence floe-size distribution and depending on conditions favoring / inhibiting
herding of ice floes into bands.

## 3    Results

In the following, we present and discuss results obtained in the Northern and Southern Hemisphere. We preferred to

not merge the results of Landsat-5 and Landsat-8 in the Northern Hemisphere because with that we have a relatively natural
discrimination between cased dominated by first-year ice (Landsat-5) and cases dominated by mixed first-year / multiyear ice
or multiyear ice (Landsat-8) (see Fig. 1).

### 3.1    Northern Hemisphere

Out of the ten products, SICCI-25km, SICCI-50km, ASI-SSMI, and SICCI-12km offer the best linear agreement with

Landsat SIC for first-year ice dominated cases as expressed, e.g., by the location of mean and median PMW SIC (red symbols)
in Fig. 2 and the values of slope, intercept and correlation coefficient listed in Table 5. The two CBT products, NOAA-CDR
and NT2-AMSRE have the smallest overall mean difference and zero median (Table 5). These four products exhibit, however,
a considerable tail of near-100 % PMW SIC values stretching across almost the entire Landsat SIC range, pointing towards
over-estimation of Landsat SIC. ASI-SSMI and NT1-SSMI SIC spread towards comparably low values at high Landsat SIC,
i.e. along the vertical dashed line denoting 100 % Landsat SIC (Fig. 2 h, i) contributing to the overall largest underestimation
of Landsat SIC among the ten products (Table 5).

For cases with mixed first-year / multiyear or multiyear ice, SICCI-25km and SICCI-50km offer best linear agreement

with Landsat SIC in the Northern Hemisphere (Fig. 3). Most other products have a less convincing linear relationship with the
majority of the data pairs being located either above (NT2-AMSR2) or below (ASI-SSMI) the identity line or within a point
cloud across this line (SICCI-12km, OSI-450, NT1-SSMI). Like for first-year ice, the two CBT products, NOAA-CDR and
NT2-AMSR have the smallest mean difference for mixed first-year / multiyear or multiyear ice (Fig. 3, Table 6). However,
particularly at higher Landsat SIC these products show many data pairs above the identity line and the linear regressions
through the mean and median PMW SIC (red dashed and solid lines) are also located above the identity line – in contrast to,
e.g. SICCI-25km and SICCI-50km.



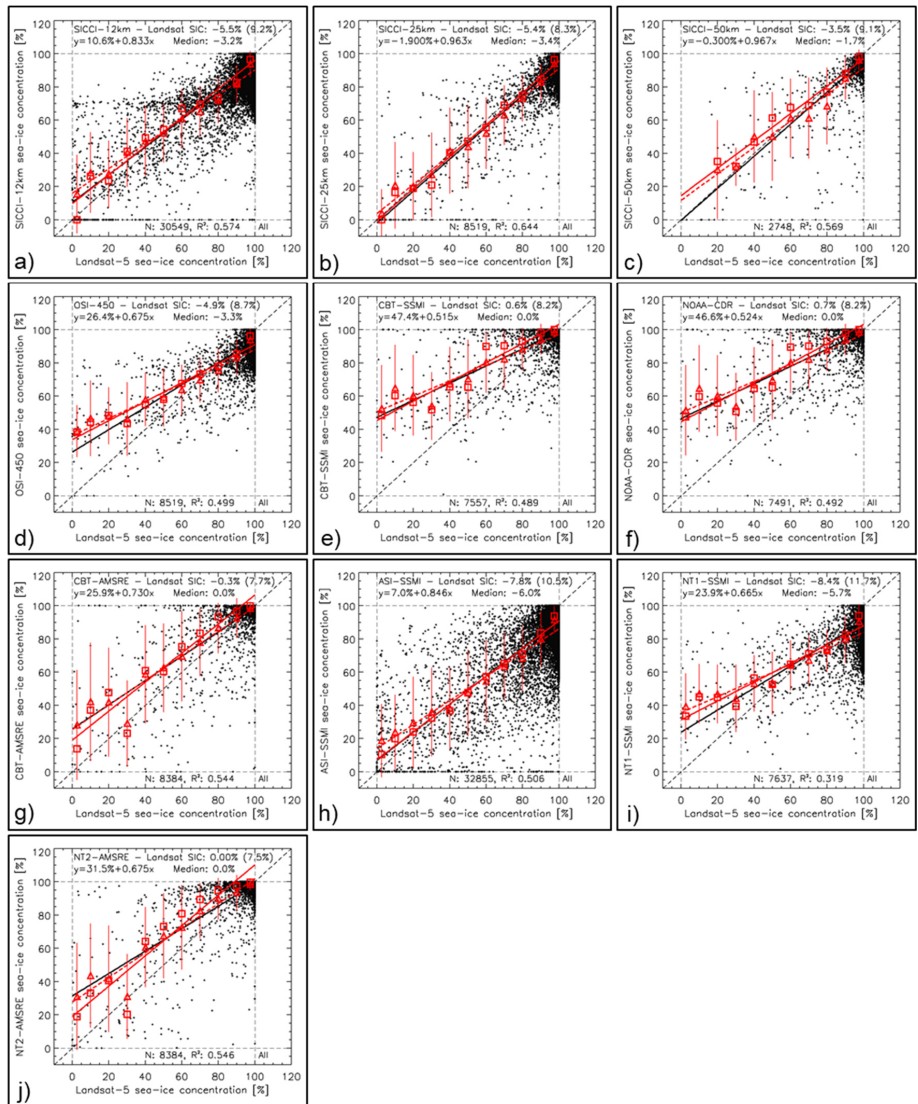

**Figure 2.** Scatterplots of PMW SIC (y-axis) versus Landsat SIC (x-axis) for all ten products for the first-year ice dominated cases from 2003-2011 in the Northern Hemisphere (Landsat-5). Black dots are individual data pairs, the black solid line is the linear regression, and the black dashed line is the identity line. Red triangles denote the mean PMW SIC computed for Landsat SIC ranges 0%-5%, 5%-15%, 15%-25%, … , 85%-95%, 95%-100%, red bars one standard deviation of these mean values and the red dashed line is the respective linear regression line. Red squares denote the median PMW SIC for the same Landsat SIC ranges and the red solid line is the respective linear regression line. The overall mean and median difference PMW SIC minus Landsat SIC, its standard deviation, and the equation of the linear regression through the individual data pairs is shown at the top, the number N of data pairs and the squared linear correlation coefficient at the bottom of each panel.



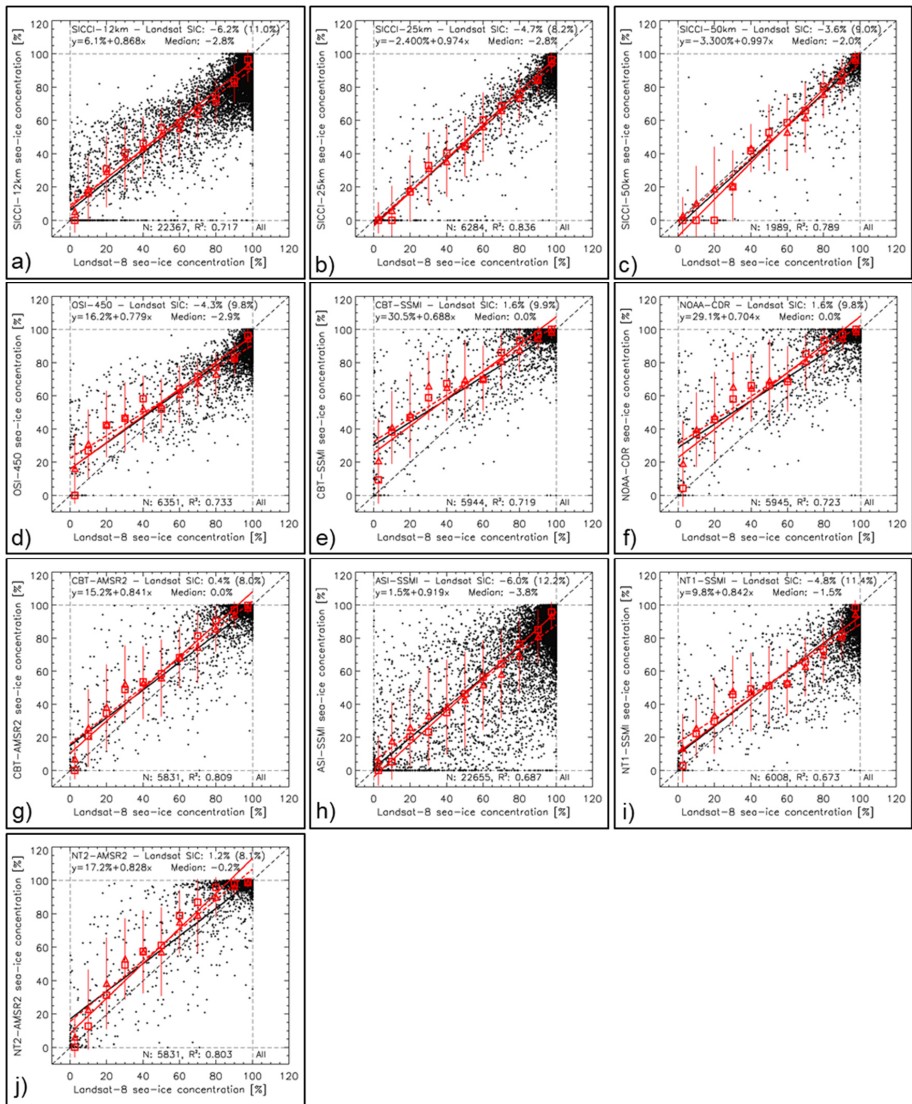

**Figure 3.** Scatterplots of PMW SIC (y-axis) versus Landsat SIC (x-axis) for all ten products for mixed first-year / multiyear or multiyear ice cases from 2013-2015 in the Northern Hemisphere (Landsat-8). See Fig. 2 for a description of symbols, lines and text.

The linear agreement between PMW SIC and Landsat SIC improves in general for all ten products for mixed first-year / multiyear or multiyear ice cases (Fig. 3, Table 6) compared to first-year ice (Fig. 2, Table 5). This improvement is comparably large for OSI-450: slope increases by ~0.10 and NT2-AMSR: slope increases by ~0.15 but quite small for SICCI-25km and SICCI-50km because slopes are close to unity already. Hence, despite the larger magnitude of overall mean and median SIC differences, of all ten products SICCI-25km and SICCI-50km provide the most stable linear agreement with Landsat SIC in the Northern Hemisphere. These two products provide SIC estimates for first-year ice which are almost as accurate as the SIC estimates for mixed first-year ice / multiyear ice or multiyear ice. This could be one consequence of the self-optimizing hybrid SICCI-2 / OSI-450 algorithm (Lavergne et al., 2019) and of the way ice tie points are chosen in comparison to the other products (e.g., Kern et al. 2020).





**Table 5.** Summary of the statistical parameters displayed in Fig. 2. Diff, DiffSDEV, and Median (all in percent SIC) are the
mean difference PMW SIC minus Landsat SIC, its standard deviation and the median difference; Slope and Intercept (in
percent SIC) are the coefficients of the linear regression, and $R^2$ and N are the squared linear correlation coefficient and number
of data pairs, respectively. Numbers in **bold** and ***bold italic*** font denote the respective "best" and "2nd best" value, respectively,
e.g. largest and 2nd-largest values of $R^2$ and lowest and 2nd-lowest values of Diff, Intercept and difference unity minus slope.

| LS5, NH 2003-11 | SICCI-12 | SICCI-25 | SICCI-50 | OSI-450 | CBT-SSMI | NOAA-CDR | CBT-AMSRE | NT1-SSMI | ASI-SSMI | NT2-AMSRE |
|---|---|---|---|---|---|---|---|---|---|---|
| Diff | -5.5 | -5.4 | -3.5 | -4.9 | 0.6 | 0.7 | *-0.3* | -8.4 | -7.8 | **0.0** |
| DiffSDEV | 9.2 | 8.3 | 9.1 | 8.7 | 8.2 | 8.2 | *7.7* | 11.7 | 10.5 | **7.5** |
| Median | -3.2 | -3.4 | -1.7 | -3.3 | 0.0 | 0.0 | 0.0 | -5.7 | -6.0 | 0.0 |
| Slope | 0.833 | *0.963* | **0.967** | 0.675 | 0.515 | 0.524 | 0.730 | 0.665 | 0.846 | 0.675 |
| Intercept | 10.6 | *-1.9* | **-0.3** | 26.4 | 47.4 | 46.6 | 25.9 | 23.9 | 7.0 | 31.5 |
| R² | *0.57* | **0.64** | 0.57 | 0.50 | 0.49 | 0.49 | 0.54 | 0.32 | 0.51 | 0.55 |
| N | 30549 | 8519 | 2748 | 8519 | 7557 | 7491 | 8384 | 7637 | 32855 | 8384 |


**Table 6.** Summary of statistical parameters shown in Fig. 3. See Table 5 for an explanation of the parameters given.

| LS8, NH 2013-15 | SICCI-12 | SICCI-25 | SICCI-50 | OSI-450 | CBT-SSMI | NOAA-CDR | CBT-AMSR2 | NT1-SSMI | ASI-SSMI | NT2-AMSR2 |
|---|---|---|---|---|---|---|---|---|---|---|
| Diff | -6.2 | -4.7 | -3.6 | -4.3 | 1.6 | 1.6 | **0.4** | -4.8 | -6.0 | *1.2* |
| DiffSDEV | 11.0 | 8.2 | 9.0 | 9.8 | 9.9 | 9.8 | **8.0** | 11.4 | 12.2 | *8.1* |
| Median | -2.8 | -2.8 | -2.0 | -2.9 | 0.0 | 0.0 | 0.0 | -1.5 | -3.8 | -1.5 |
| Slope | 0.868 | *0.974* | **0.997** | 0.779 | 0.688 | 0.704 | 0.841 | 0.842 | 0.919 | 0.828 |
| Intercept | 6.1 | *-2.4* | -3.3 | 16.2 | 30.5 | 29.1 | 15.2 | 9.8 | **1.5** | 17.2 |
| R² | 0.72 | **0.84** | 0.79 | 0.73 | 0.72 | 0.72 | *0.81* | 0.67 | 0.69 | 0.80 |
| N | 23433 | 6484 | 2056 | 6576 | 5944 | 5945 | 5831 | 6008 | 22655 | 5831 |


### 3.2  Southern Hemisphere
In the Southern Hemisphere, slope and location of the linear regression lines as well as of the mean and median PMW
SIC values (red symbols) is more similar between the ten products (Fig. 4, Table 7). The linear agreement is fairly good for
SICCI-2 products, CBT-AMSR2 and ASI-SSMI. Like in the Northern Hemisphere, SICCI-25km and SICCI-50 km reveal the
best linear agreement with Landsat SIC but SICCI-50km appears to be negatively biased. This bias is associated with a large
number of PMW SIC values of 0 % at non-zero Landsat SIC which is also reflected by the mean and median PMW SIC
(compare Fig. 4c) with Fig. 3c)). We discuss this issue and the observation that all products except CBT-SSMI, NOAA-CDR
and CBT-AMSR2 exhibit SIC values below about 10-15 % while these three products lack values in the PMW SIC range
between 0 % and ~15 % in Section 5.4.
Like in the Northern Hemisphere (Table 6), the magnitude of the SIC difference is smallest for NT2-AMSR2, NOAA-
CDR and the two CBT products and largest for NT1-SSMI and ASI-SSMI. Of all ten products, NT2-AMSR2 (Fig. 4 j) offers
the most asymmetric SIC distribution and a considerable overestimation of Landsat SIC in the range between ~40 % and ~90
%, also expressed by median SIC > mean SIC for all Landsat SIC bins above 25 % (Fig. 4 j). NT2-AMSR2 is the only product
with a substantial positive overall mean difference of 3.4 %, even the median difference is > 0 % (Table 7).



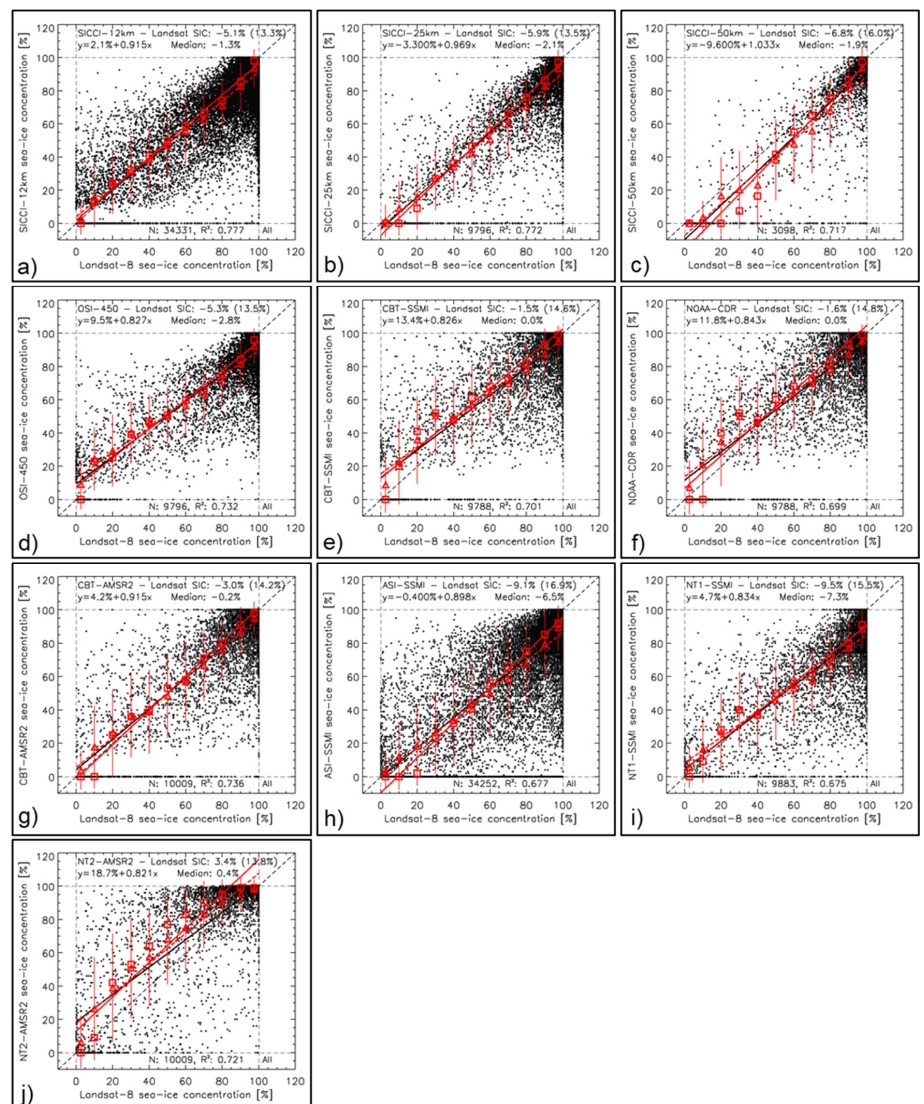

**Figure 4.** Scatterplots of PMW SIC (y-axis) versus Landsat SIC (x-axis) for all ten products for 2013-2015 in the Southern Hemisphere. See Fig. 2 for a description of symbols, lines and text.

**Table 7.** Summary of statistical parameters shown in Fig. 4. See Table 5 for an explanation of the parameters given.

| LS8, SH 2013-15 | SICCI-12 | SICCI-25 | SICCI-50 | OSI-450 | CBT-SSMI | NOAA-CDR | CBT-AMSR2 | NT1-SSMI | ASI-SSMI | NT2-AMSR2 |
|---|---|---|---|---|---|---|---|---|---|---|
| Diff | -5.1 | -5.9 | -6.8 | -5.3 | **-1.5** | ***-1.6*** | -3.0 | -9.5 | -9.1 | 3.4 |
| DiffSDEV | **13.3** | ***13.5*** | 16.0 | ***13.5*** | 14.6 | 14.8 | 14.2 | 15.5 | 16.9 | 13.8 |
| Median | -1.3 | -2.1 | -1.9 | -2.8 | 0.0 | 0.0 | -0.2 | -7.3 | -6.5 | 0.4 |
| Slope | 0.915 | **0.969** | ***1.033*** | 0.827 | 0.826 | 0.843 | 0.915 | 0.834 | 0.898 | 0.821 |
| Intercept | ***2.1*** | -3.3 | -9.6 | 9.5 | 13.4 | 11.8 | 4.2 | 4.7 | **-0.4** | 18.7 |
| R² | **0.78** | ***0.77*** | 0.72 | 0.73 | 0.70 | 0.70 | 0.74 | 0.68 | 0.68 | 0.72 |
| N | 34331 | 9796 | 3098 | 9796 | 9788 | 9788 | 10009 | 9883 | 34252 | 10009 |





### 3.3 Hemispheric Similarities and Differences

Overall, agreement between PMW SIC and Landsat SIC differs between the two hemispheres. For all products, we find a substantially larger scatter of SIC values around the identity line in the Southern Hemisphere (section 3.2) than the Northern Hemisphere (section 3.1). One the one hand, this larger scatter in the Southern Hemisphere could be the result of a considerably larger number of Landsat scenes of cases with low SIC, naturally resulting in a larger spread of the SIC. In addition, the majority of the Landsat scenes in the Southern Hemisphere reflect late spring / summer conditions. During such conditions, snow metamorphism due to melt and melt-refreeze cycles substantially change the sea ice surface emissivity on daily time-scales and sub grid-cell size spatial scales (e.g. Willmes et al., 2014) causing a larger scatter in SIC. On the other hand, we are dealing with an unknown amount of overestimation of the actual sea-ice concentration by our Landsat SIC during summer melt due to mixed ocean-ice Landsat pixels (Subsection 2.2.4). We refer to Sections 4.3, 5.1 and 5.2 for more discussion on this issue.

In general, we find the scatter is larger for products offered at finer grid resolution, e.g. SICCI-12km and ASI-SSMI, than for the coarser grid-resolution products. The larger number of valid SIC pairs of the high-resolution products result in more scatter due to the inherent retrieval noise – even though the capability to resolve smaller-scale SIC variations is better for the fine- than the coarser-resolution products (see section 5.1). In addition, a mismatch in the location of a, for example, 10km-scale ice tongue between a Landsat scene and a PMW SIC product has a substantially larger influence on the SIC difference at 12.5 km than at 25 or 50 km grid resolution. The fact that oversampling is much larger at 12.5 km than at 50 km plays a role here also. Even using simulated brightness temperatures one gets a large spread between a reference SIC and the PMW SIC due to resolution mismatch (see e.g. Tonboe et al., 2016). Note in this context that we estimate Landsat SIC at the grid resolution of the respective products, i.e. 12.5 km, 25.0 km or 50.0 km.

**Table 8.** Comparison of statistical parameters listed in Tables 5 and 6 in the Northern Hemisphere for SICCI-2 and OSI-450 products using truncated or non-truncated (near-100 % SIC) PMW SIC data. See Table 5 for an explanation of the parameters given. Top (LS5, NH 2003-11) is for first-year ice dominated cases, bottom (LS8, NH 2013-15) is for mixed first-year / multiyear and multiyear ice cases. The overall median differences do not change and are not listed again.

| LS5, NH 2003-11 | SICCI-12 | SICCI-12 non-truncated | SICCI-25 | SICCI-25 non-truncated | SICCI-50 | SICCI-50 non-truncated | OSI-450 | OSI-450 non-truncated |
|---|---|---|---|---|---|---|---|---|
| Diff | -5.5 | -4.6 | -5.4 | -5.0 | -3.5 | -3.0 | -4.9 | -4.5 |
| DiffSDEV | 9.2 | 10.0 | 8.3 | 8.7 | 9.1 | 9.3 | 8.7 | 9.0 |
| Slope | 0.833 | 0.852 | 0.963 | 0.974 | 0.967 | 0.979 | 0.675 | 0.684 |
| Intercept | 10.6 | 9.6 | -1.9 | -2.5 | -0.3 | -1.0 | 26.4 | 26.0 |
| R² | 0.57 | 0.54 | 0.64 | 0.63 | 0.57 | 0.56 | 0.50 | 0.48 |
| LS8, NH 2013-15 | | | | | | | | |
| Diff | -6.2 | -4.9 | -4.7 | -4.4 | -3.6 | -3.4 | -4.3 | -3.9 |
| DiffSDEV | 11.0 | 12.1 | 8.2 | 8.5 | 9.0 | 9.1 | 9.8 | 9.9 |
| Slope | 0.868 | 0.891 | 0.974 | 0.982 | 0.997 | 1.000 | 0.779 | 0.786 |
| Intercept | 6.1 | 5.2 | -2.4 | -2.7 | -3.3 | -3.5 | 16.2 | 15.9 |
| R² | 0.72 | 0.68 | 0.84 | 0.83 | 0.79 | 0.79 | 0.73 | 0.73 |

SICCI-2 products and OSI-450 provide access to SIC values above 100 % and below 0 % that are naturally retrieved due to the brightness temperature distribution around ice and water tie points used. Kern et al. (2019) found that incorporation of these so-called off-range or non-truncated SIC values provides a more accurate estimate of accuracy, i.e. difference to the true SIC value, and precision, i.e. standard deviation of this difference. Table 8 reveals that independent of the ice type, the



magnitude of the mean difference decreases while the slope of the linear regression increases, becoming closer to unity, in
agreement to Kern et al. (2019). We observe the same in the Southern Hemisphere (Table S05 in the supplementary material).
Of particular interest in this regard are high-concentration cases discussed in more detail in Section 4.2 but also the effect of
the truncation at 0 % in the context of filters used to mitigate spurious SIC values (see Section 5.3).

## 4    Case Studies

In the previous section, we showed results independent of the ice regime (see below) – except for some general
discussion about the observed differences between cases with predominantly first-year ice (Landsat-5) and cases with a mixture
of first-year / multiyear or multiyear ice (Landsat-8). This section deals with our comparison between PMW SIC and Landsat
SIC for the following ice regimes: "ice edge", "leads and openings" = cases with leads and coastal polynyas or openings,
"heterogeneous ice" = cases with irregularly shaped openings in the ice pack, "freeze-up", "high-concentration ice", and "melt
conditions" (see Table S01 in the supplementary material). We show in more detail results of the last three ice regimes known
to cause biases in PMW SIC products. For all remaining regimes we show examples in Figs. S04 through S09 in the
supplementary material while the results of the statistical comparison for all regimes will be summarized in Figs 10 and 11.

### 4.1    Freeze-Up

These are cases where according to the date, geographic location and information in the Landsat scene freeze-up has
commenced. We select Landsat scenes containing a considerable fraction of new and thin ice; these are acquired in September
and February/March in the Northern and Southern Hemisphere, respectively. We have got only few such cases in both
hemispheres (see Table S01 in the supplementary material). We expect PMW SIC underestimates Landsat SIC (LSIC) –
particularly for young and thin ice with a thickness < 0.2 m (e.g. Ivanova et al., 2015). Figure 5 illustrates the conditions for a
Landsat-8 scene close to Greenland in the Fram Strait on September 15 2015. The classified Landsat-8 image (Fig. 5, top left)
reveals a mix of large ice floes – presumably second-year or older ice – and meandering patches of smaller floes embedded
into a matrix of mostly grey and a few dark pixels; the grey pixels are supposed to represent bare / thin sea ice, the dark pixels
open water. All products agree well with Landsat SIC in the topmost part of the scene over high-concentration ice. PMW SIC
maps of six of the ten products (SICCI-2 products, OSI-450, NT1-SSMI and ASI-SSMI) reveal an overall SIC distribution
similar to Landsat SIC. For the remaining four products, the SIC difference maps show widespread overestimation of LSIC by
PMW SIC expressed by positive (red) values. Unlike expected, we do not observe negative SIC differences for the entire
greyish area of the Landsat-8 scene.
The main reason for this observation is the actual ice condition. Very likely the greyish area represents a mixture of
sub-pixel size, i.e. less than 30 m x 30 m, ice floes and brash ice formed from disintegrated thicker ice floes and young / new
sea ice. On the one hand, the sub-pixel size floes and the brash ice are thicker than young / new sea ice. These forms of sea ice
exhibit different surface properties and hence microwave emissivity than young / new thin sea ice. For such a mixture of ice
types, it is particularly difficult to retrieve an accurate SIC with any of the algorithms used in the ten products. Ice tie points
do not adequately represent these ice conditions. On the other hand, for the greyish area the Landsat SIC could likely be too
high because of mixed ocean-ice Landsat pixels (see Subsection 2.2.4 and the respective supplementary material). Hence, what
appears to be 100 % thin ice might be just 50 % thick ice. However, observations conducted at Henrik Krøyer Holme station
(80°38'N 13°43'W, see star in Fig. 5, top left panel) on September 15 2015 and the preceding days indicate freezing conditions
with air temperatures between -5°C and -13°C (https://www.dmk.dk/vejrarkiv, last access: June 29 2021). Therefore, it is quite
likely, new / thin ice covers most open water patches and any over-estimation of Landsat SIC due to sub-pixel size open water



patches is rather small. Therefore, to our opinion, the observed differences PMW SIC minus Landsat SIC are mainly caused
by the above-mentioned difficulties of the PMW SIC algorithms to handle the complex sea-ice conditions encountered.

**Figure 5.** Landsat SIC, PMW SIC, and the difference PMW SIC minus Landsat SIC (LSIC) for all ten products for a freeze-
up scene in the Fram Strait on September 15, 2015. The Landsat surface class map at the top left shows white: thick / snow-
covered ice; grey: bare / thin ice; black: open water). The red star marks the location of Henrik Krøyer Holme station (see
text). White and grey pixels are used to compute maps of gridded LSIC at 12.5 km, 25 km and 50 km, respectively (blue:
outside Landsat image). A subset of SICCI-12km SIC grid cells shown at the top right illustrates the array used for the
collocation. Panels in the remaining four rows show PMW SIC and PMW SIC minus LSIC for all ten products. Land is flagged
brown in the SIC panels and black in the SIC difference panels; it differs between the PMW products. The land masks in the
two bigger maps at the top come from the plotting routine used. LSIC maps use the land masks of the SICCI-2 products.


Figure 6 illustrates a freeze-up case in Pine Island Bay, Amundsen Sea, Southern Ocean, on March 12, 2014. The

classified Landsat-8 scene features a predominant coverage with new / young ice, some open water towards the coast and little
thick / snow covered ice and icebergs in the open bay. Landsat SIC is mostly around 90 %; only few grid cells with low SIC
exist close to the coast at 12.5 km and 25 km grid resolution. Nine of the ten PMW SIC products reveal considerably lower
SIC values with SICCI-25km, OSI-450, NT1-SSMI and ASI-SSMI exhibiting particularly large widespread negative
differences with magnitude > 40 %. An exception is NT2-AMSR2 exhibiting the highest PMW SIC of all ten products and





overall the smallest differences. It is the only product, though, which also exhibits positive differences, i.e. an over-estimation
of Landsat SIC by up to 20 %.
The widespread under-estimation of Landsat SIC by almost all products is very well in line with the findings of
Ivanova et al. (2015), albeit a bit large in magnitude. The new ice encountered in our example comprises a comparably large
fraction of frazil / grease / small pancake ice while compared to nilas and grey/grey-white ice in Ivanova et al. (2015). Because
Pine Island Glacier Automatic Weather Station (see star in top left map of Fig. 6) reported air temperatures between -11°C and
-21°C on March 12, 2014 and the three preceding days (Mojica Moncada et al., 2019), the grey pixels in this Landsat scene
very likely represent new/thin sea ice formed locally. However, we cannot fully exclude an over-estimation of Landsat SIC by
sub-pixel size open water patches between streaks of new ice formed being classified as thin ice instead of open water (see
Subsection 2.2.4 and respective supplementary material); for the conditions encountered this positive bias in Landsat SIC
should be less than 5 %, maximum 10 %. The existence of such a positive bias combined with the different ice type encountered
compared to Ivanova et al. (2015) explains why the observed under-estimation of Landsat SIC for most of the PMW SIC
products is larger in magnitude than expected.

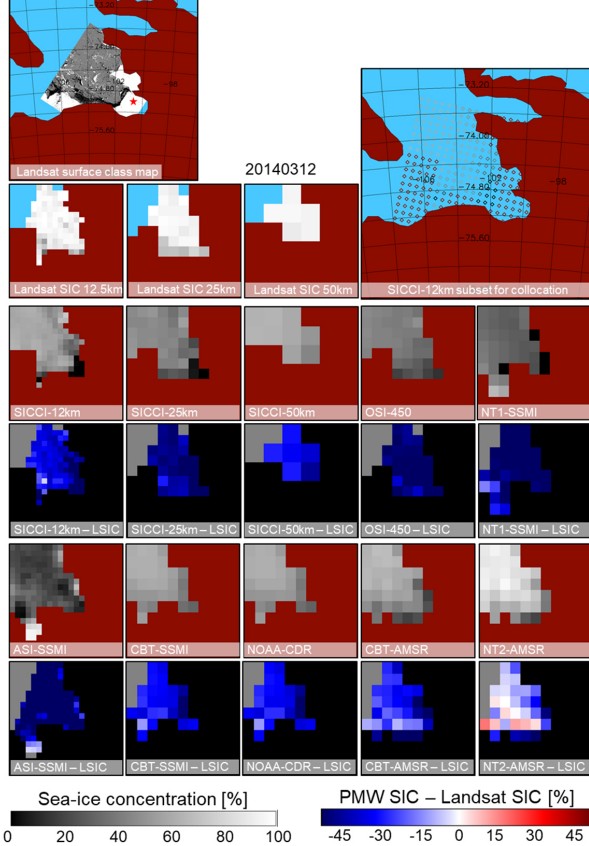


**Figure 6.** Landsat SIC, PMW SIC, and the difference PMW SIC minus Landsat SIC for all ten products for a scene near the
coast during freeze-up in Pine Island Bay, Amundsen Sea, Southern Ocean, on March 12, 2014. The red star in the top left
map marks the location of the Pine Island Glacier Automatic Weather Station (see text). Some of the white patches near the
coast in this map are actually glacier ice not adequately flagged by the land mask. See Fig. 5 for more details.





Table 9 summarizes our results of the freeze-up cases for which we expected, overall, an under-estimation of Landsat
SIC, i.e. a negative bias, due to a notable fraction of new / thin ice (see Ivanova et al., 2015). In the Northern Hemisphere,
performance of the products differs a lot. We find positive biases for the two CBT-products, NOAA-CDR and NT2-AMSR2,
large negative biases for the remaining products. SICCI-25km offers the best linear agreement with Landsat SIC. In the
Southern Hemisphere, a number of products have a regression line slope close to unity, a small intercept and a squared linear
correlation coefficient > 0.8. Most importantly, however, all products – except NT2-AMSR2 – on average under-estimate
Landsat SIC in agreement with Ivanova et al. (2015).

**Table 9.** Summary of statistical results obtained for three freeze-up cases in the Northern Hemisphere (NH) and for 11 freeze-
up cases in the Southern Hemisphere (SH) using Landsat 8 data. See Table 5 for an explanation of the parameters given.

| NH | SICCI-12 | SICCI-25 | SICCI-50 | OSI-450 | CBT-SSMI | NOAA-CDR | CBT-AMSR2 | NT1-SSMI | ASI-SSMI | NT2-AMSR2 |
|---|---|---|---|---|---|---|---|---|---|---|
| Diff | -8.2 | -8.9 | -10.5 | -7.7 | 5.0 | 4.6 | **2.6** | -14.1 | -12.0 | *4.3* |
| Diff SDEV | 13.5 | **10.8** | 17.8 | 13.9 | 18.5 | 18.4 | *12.9* | 20.8 | 21.9 | 13.8 |
| Slope | 0.799 | **0.960** | *0.948* | 0.665 | 0.655 | 0.679 | 0.881 | 0.673 | 0.738 | 0.866 |
| Intercept | 7.8 | **-5.7** | *-6.4* | 19.3 | 31.6 | 29.4 | 12.0 | 11.3 | 8.6 | 14.9 |
| R² | *0.77* | **0.84** | 0.65 | 0.70 | 0.58 | 0.58 | *0.77* | 0.50 | 0.51 | 0.74 |
| N | 751 | 208 | 64 | 210 | 191 | 191 | 186 | 196 | 702 | 186 |
| SH |  |  |  |  |  |  |  |  |  |  |
| Diff | -11.8 | -12.1 | -7.4 | -12.1 | -6.3 | *-6.1* | -6.5 | -10.9 | -11.4 | **2.1** |
| Diff SDEV | 18.1 | 15.9 | 16.1 | 15.1 | 12.1 | 12.1 | *11.8* | 15.3 | 18.1 | **10.6** |
| Slope | 0.839 | 0.915 | *1.027* | 0.861 | 0.965 | 0.971 | 0.977 | 0.953 | **0.982** | 0.943 |
| Intercept | *2.0* | -4.8 | -9.7 | **0.1** | -3.3 | -3.7 | -4.5 | -6.9 | -9.8 | 7.0 |
| R² | 0.66 | 0.72 | 0.75 | 0.73 | 0.83 | *0.84* | *0.84* | 0.75 | 0.72 | **0.86** |
| N | 1843 | 531 | 169 | 531 | 536 | 536 | 547 | 540 | 1842 | 547 |

**4.2      High-concentration ice**
These are cases where the Landsat scene indicates either a closed ice cover without any leads or openings or an almost
closed ice cover with few refrozen leads or openings, resulting in near-100 % Landsat SIC. In the ideal case, we expect PMW
SIC is close to 100 %. Figure 7 illustrates such a case for April 4, 2015 in the Beaufort Sea, Arctic Ocean. Landsat SIC is
100.0 %. All ten PMW SIC products exhibit quite high sea-ice concentrations – particularly SICCI-50km, NOAA-CDR and
NT2-AMSR2. However, the difference maps clearly reveal a (very) small and negative bias for all PMW products. This bias
is largest in magnitude for SICCI-12km and ASI-SSMI and smallest in magnitude for NT2-AMSR2.
Table 10 summarizes the results obtained for, in total, 40 high-concentration cases in the Northern Hemisphere: 28
first-year ice dominated scenes (Landsat-5) and 12 scenes of mixed first-year / multiyear or multiyear ice cases (Landsat-8).
We find the largest biases for SICCI-12km and ASI-SSMI independent of ice type. Except for CBT-AMSR and NT2-AMSR,
all products exhibit a larger bias for first-year ice cases than mixed first-year / multiyear or multiyear ice cases. We hypothesize
that the different biases between PMW and Landsat SIC for these near-100 % cases are caused by the different capabilities of
the respective algorithms to derive an accurate SIC independent of ice type – as stated already in Section 3.1. NT1-SSMI and
ASI-SSMI appear to have the largest difficulties with the different ice types encountered because their biases vary most. We
note the two CBT products and NOAA-CDR (and NT2-AMSR2) have a median difference of 0.0 % independent of ice type
– similar to Tables 5 and 6. For SICCI-2 products and OSI-450, median differences are smaller in magnitude than for all ice
and approach zero for the mixed first-year / multiyear or multiyear ice cases.



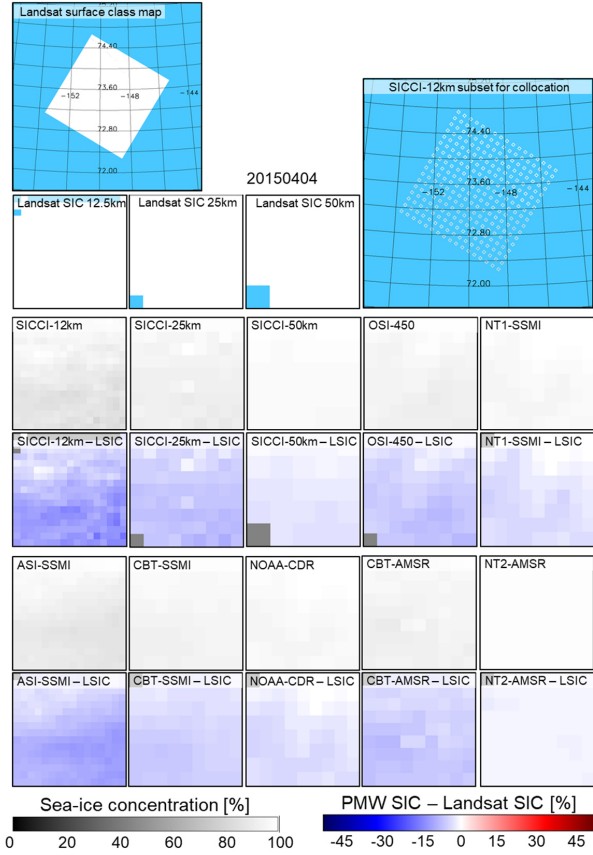

**Figure 7.** Landsat SIC, PMW SIC, and the difference PMW SIC minus Landsat SIC for all ten products for a high-concentration scene in the Beaufort Sea, Arctic Ocean, on April 4, 2015. See Fig. 5 for a description of the maps shown.

**Table 10.** Summary of statistical results obtained in the Northern Hemisphere for 28 cases with first-year ice (top, LS5, NH 2003-11) and for 12 cases with mixed first-year / multiyear or multiyear ice (bottom, LS8, NH 2013-15). See Table 5 for an explanation of the parameters shown. For SICCI-2 and OSI-450 products, we include in all rows but "N" behind the "/" values based on non-truncated (near 100 %) SIC data. We omit slope and intercept because SIC data pairs cluster at 100 % and do not allow a meaningful estimation of a linear regression line.

| LS5, NH 2003-11 | SICCI-12 | SICCI-25 | SICCI-50 | OSI-450 | CBT-SSMI | NOAA-CDR | CBT-AMSR | NT1-SSMI | ASI-SSMI | NT2-AMSR |
|---|---|---|---|---|---|---|---|---|---|---|
| Diff | -4.0 / -3.0 | -3.7 / -3.4 | -1.5 / -1.0 | -3.5 / -3.2 | -0.8 | -0.7 | -0.9 | -5.8 | -6.9 | -0.6 |
| DiffSDEV | 5.2 / 6.0 | 4.0 / 4.4 | 1.8 / 2.5 | 3.7 / 4.1 | 1.6 | 1.4 | 1.8 | 6.6 | 5.6 | 1.4 |
| Median | -2.6 / -2.6 | -2.5 / -2.5 | -1.0 / -1.0 | -2.4 / -2.4 | 0.0 | 0.0 | 0.0 | -3.5 | -6.0 | 0.0 |
| N | 7028 | 1978 | 677 | 1978 | 1940 | 1940 | 2104 | 1940 | 7633 | 2104 |
| LS8, NH 2013-15 | | | | | | | | | | |
| Diff | -2.9 / -0.8 | -1.5 / -0.5 | -0.9 / -0.4 | -1.3 / -0.3 | -0.5 | -0.2 | -1.0 | -0.3 | -2.6 | -0.6 |
| DiffSDEV | 4.1 / 6.2 | 2.2 / 3.1 | 1.2 / 1.7 | 1.9 / 3.0 | 1.4 | 0.9 | 3.0 | 0.9 | 2.6 | 2.5 |
| Median | -0.2 / -0.2 | -0.2 / -0.2 | -0.3 / -0.3 | -0.2 / -0.2 | 0.0 | 0.0 | 0.0 | 0.0 | -2.1 | -0.5 |
| N | 2659 | 764 | 242 | 764 | 714 | 714 | 723 | 714 | 2571 | 723 |

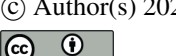
Using non-truncated SIC of SICCI-2 products and OSI-450 (see also Table 8), reduces the magnitude of the bias by

between 0.5 % for SICCI-50km and 2.1 % for SICCI-12km for the mixed first-year / multiyear or multiyear ice cases (LS8)

and less than that for the first-year ice cases. As expected, the standard deviation of the bias increases using non-truncated SIC.

The other six PMW products set SIC values > 100 % to 100 % or do not permit a simple retrieval of such SIC values (NT2-

AMSR2, but see Ivanova et al., 2015), and would therefore have a different bias and a larger standard deviation than shown in

Table 10 (see Kern et al., 2019). Of the SICCI-2 / OSI-450 products, SICCI-50km provides the smallest bias and the smallest

standard deviation of this bias: -0.7 % ± 2.2 % in line with Kern et al. (2019) who reported a bias of -0.5 % ± 2.1 % for non-

truncated SICCI-50km SIC. The median difference of SICCI-2 products and OSI-450 is quite similar to the mean difference

using non-truncated SIC for mixed first-year / multiyear or multiyear ice cases.

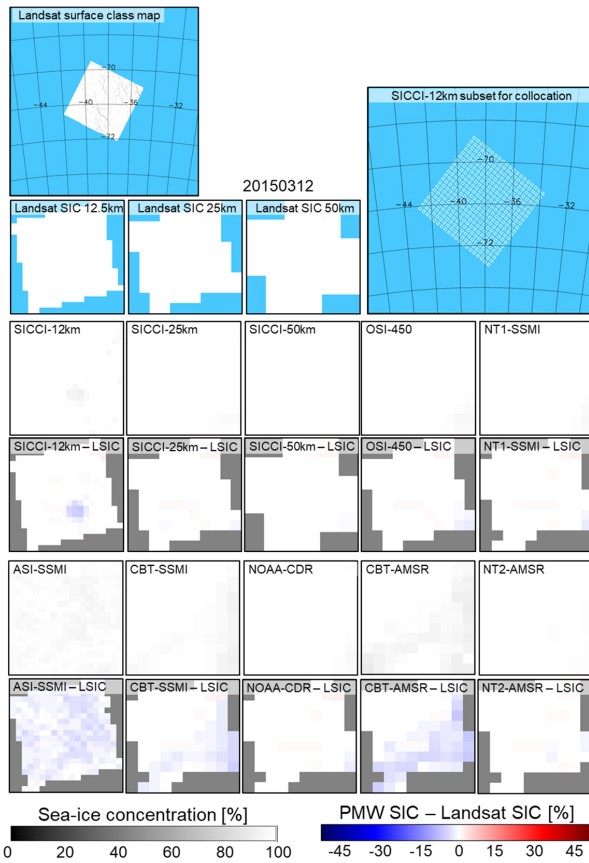


**Figure 8.** Landsat SIC, PMW SIC, and the difference PMW SIC minus Landsat SIC for all ten products for a high-
concentration scene in the Weddell Sea, Southern Ocean, on March 12, 2015. See Fig. 5 for a description of the maps shown.

Figure 8 illustrates a high-concentration case in the Weddell Sea, Southern Ocean, on March 12, 2015. Six of the ten

PMW SIC products show almost 100 % sea-ice concentration and almost zero bias. We only find notable deviations from 100
% concomitant with a small negative bias for ASI-SSMI, the two CBT-products and SICCI-12km. For our four high-
concentration cases in the Southern Ocean (Table 11), we find the largest overall bias for ASI-SSMI. Most products reveal a
bias of magnitude 0.3 % or smaller.




Using non-truncated SICCI-2 and OSI-450 SIC results in positive biases, ranging between 1.8 % for OSI-450 and 2.7
% for SICCI-50km (Table 11, values behind the "/"). This amounts to an increase of the mean SICCI-2 / OSI-450 SIC for
these cases by ~ 2.5 %. This increase is larger than in the Northern Hemisphere (compare Table 10). We explain this with a
comparably large fraction of PMW SIC > 100 % for our small high-concentration cases data set of the Southern Hemisphere
(four) compared to the Northern Hemisphere (40 in total). This is confirmed by median differences increasing from near-0 %
to about the value of the mean differences using non-truncated SIC (Table 11).
Three of the four high-concentration cases identified in the Southern Ocean are from months November / December,
a time of the year when melt onset and melt-refreeze cycles cause higher variability of the ice emissivity. One of the likely
impacts is a notable fraction of PMW SIC > 100 % (see Fig. S01 in the supplementary material). The same applies in a different
way to the case shown in Fig. 8, the only late fall / early winter case out of these four cases. The overall Landsat SIC of this
scene is 99.9 %; that of an adjacent scene is 98.9 % (not shown). Sea ice and snow properties in late fall / early winter are often
as well quite variable and can cause an elevated spread of the retrieved PMW SIC around 100 %, resulting in a substantial
fraction of SIC values > 100 %. For instance, the overall SICCI-25km SIC is 101.9 % for the scene shown in Fig. 8 and 103.1
% for the adjacent scene (not shown).
**Table 11.** Summary of statistical results obtained for the four high concentration cases in the Southern Hemisphere. See Table
5 for an explanation of the parameters shown. For SICCI-2 and OSI-450 products, we include in rows "Diff", "DiffSDEV",
and "Median" behind the "/" values obtained using non-truncated SIC.

| LS8, SH 2013-15 | SICCI-12 | SICCI-25 | SICCI-50 | OSI-450 | CBT-SSMI | NOAA-CDR | CBT-AMSR2 | NT1-SSMI | ASI-SSMI | NT2-AMSR2 |
|---|---|---|---|---|---|---|---|---|---|---|
| Diff | -0.1 / 2.5 | 0.0 / 2.4 | 0.0 / 2.7 | -0.3 / 1.8 | -0.7 | 0.1 | -1.1 | -0.9 | -2.9 | -0.1 |
| DiffSDEV | 1.7 / 2.9 | 0.8 / 2.3 | 1.2 / 2.7 | 2.1 / 3.1 | 1.7 | 0.7 | 2.0 | 2.6 | 2.5 | 1.2 |
| Median | 0.0 / 2.8 | 0.0 / 2.5 | 0.1 / 2.6 | 0.0 / 2.2 | 0.0 | 0.1 | 0.0 | 0.0 | -2.4 | 0.0 |
| N | 978 | 287 | 93 | 287 | 288 | 288 | 302 | 288 | 973 | 302 |

### 4.3 Melt conditions

For melt-condition cases, we select Landsat scenes by means of the calendar date. In the Northern Hemisphere, we
consider the time-period May 15 to May 31, in the Southern Hemisphere we used the time-period November 15 to February
28. We did not include Landsat scenes subject to melt ponding on sea ice, e.g. during months June through August; this topic
is covered in Kern et al. (2020).
**Table 12.** Summary of statistical results obtained for 15 melt-condition cases (without melt-ponds) in the Northern
Hemisphere. See Table 5 for an explanation of the parameters shown. Numbers added behind the "/" for SICCI-2 and OSI-
450 products denote the results obtained using non-truncated SIC.

| LS8, NH 2013-15 | SICCI-12 | SICCI-25 | SICCI-50 | OSI-450 | CBT-SSMI | NOAA-CDR | CBT-AMSR2 | NT1-SSMI | ASI-SSMI | NT2-AMSR2 |
|---|---|---|---|---|---|---|---|---|---|---|
| Diff | -5.3 / -4.3 | -5.1 / -4.6 | -4.2 / -4.2 | -4.6 / -4.3 | 2.2 | 2.4 | **0.2** | -3.5 | -4.7 | *1.7* |
| DiffSDEV | 10.5 / 11.2 | 8.9 / 9.3 | 9.6 / 9.6 | 9.5 / 9.7 | 9.8 | 9.7 | **7.4** | 10.8 | 12.2 | *8.3* |
| Slope | 0.829/0.852 | **0.930/0.943** | *0.898/0.899* | 0.617/0.626 | 0.418 | 0.416 | 0.727 | 0.637 | 0.740 | 0.564 |
| Intercept | 10.5 / 9.4 | **1.4 / 0.6** | *5.3 / 5.2* | 30.9 / 30.4 | 56.9 | 57.3 | 26.1 | 30.6 | 19.5 | 43.0 |
| R² | *0.67* / 0.65 | **0.72 / 0.71** | 0.61 / 0.61 | 0.61 / 0.60 | 0.54 | 0.54 | 0.66 | 0.48 | 0.55 | 0.56 |
| N | 2926 | 817 | 266 | 817 | 817 | 817 | 795 | 823 | 3117 | 795 |





In the Northern Hemisphere (Table 12), we find positive and comparably small biases for the two CBT products,
NOAA-CDR and NT2-AMSR2, negative biases for all other products. We find the best quality of the linear agreement between
PMW SIC and Landsat SIC for SICCI-25km, followed by SICCI-50km and SICCI-12km. According to Kern et al. (2020), the
second half of May is characterized by an upswing of number and magnitude of SIC values > 100 % for SICCI-2 / OSI-450
products (see Fig. S02 in the supplementary material). Using non-truncated SIC of these products reduces the mean bias by
1.0 % for SICCI-12km, 0.5 % for SICCI-25km, and 0.3 % for OSI-450 and further improves the already good linear agreement.
For SICCI-50km, results remain almost unchanged. We explain the difference in the response between SICCI-50km and
SICCI-12km with the larger sensitivity of the higher frequency channels used by SICCI-12km to early stages of melt
encountered at that time of the year.
Figure 9 illustrates a typical case of late summer melt conditions in the Ross Sea, Southern Ocean. The classified
Landsat-8 image shows a heterogeneous mixture of black, grey and white pixels. The grey pixels denote a mixture of open
water and thicker ice, possibly brash ice, sea ice with a wet or even flooded snow cover, or bare relatively thick ice from which
the snow has been washed off. New/young ice is unlikely according to 6-hourly forecasts of the Antarctic Mesoscale Prediction
System (AMPS) revealing near-surface temperatures around -1°C on January 27 2014 and between -3°C and -5°C on January
28 and 29 2014 (http://polarmet.osu.edu/AMPS/, last access June 29, 2021), indicating that freeze-up has not yet commenced.

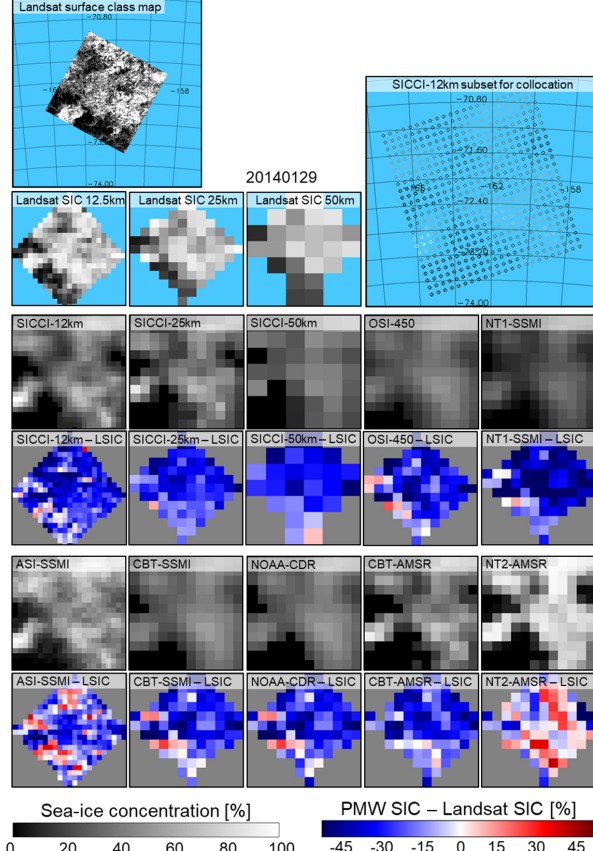


**Figure 9.** Landsat SIC, PMW SIC, and difference PMW SIC minus Landsat SIC for all ten products for a melt-condition case
in the Ross Sea, Southern Ocean, on January 29, 2014. See Fig. 5 for more description of the maps shown.




PMW SIC distributions match well with Landsat SIC, which is > 70 % for a considerable fraction of the map, but for

most products PMW SIC is considerably smaller. Negative biases dominate and are widespread 30 % to 50 % in magnitude.
Striking is the similarity between Landsat SIC 12.5km and ASI-SSMI, and between Landsat SIC 25 km and SICCI-25km as
well as CBT-AMSR2. Striking is also the similarity between OSI-450, NT1-SSMI, CBT-SSMI and NOAA-CDR. These
similarities indicate different native spatial resolutions, TB sampling intervals and grid spacings of, SSMI(S) on the one hand
and AMSR-E(2) on the other hand, can cause a substantial difference in the agreement with independent SIC estimates such
as from Landsat – especially when ice conditions are as heterogeneous as in this example (see Section 5.1). There is a notable
fraction of positive biases, e.g. for ASI-SSMI – LSIC (see also Fig. S03 in the supplementary material) and NT2-AMSR –
LSIC. NT2-AMSR – LSIC even tends to show more grid cells with positive than negative biases – not just in this case.

Overall, we find negative biases for nine of the ten products in the Southern Hemisphere (Table 13). These are smallest

in magnitude for CBT-SSMI and NOAA-CDR: < 1 %, and largest for NT1-SSMI, ASI-SSMI and SICCI-50km. NT2-AMSR2
stands out as the only product with a positive bias of 5 % (see Section 5.2). SICCI-25km and SICCI-50km again provide the
best linear agreement with Landsat SIC (Table 13). Results for SICCI-2 products and OSI-450 improve when using non-
truncated SIC (see also Fig. S01 in the supplementary material). In contrast to the Northern Hemisphere (see Table 12, Fig.
S02 in the supplementary material), also SICCI-50km reveals a reduction of the bias and increase in the linear regression line
slope. We attribute this to i) the presence of advanced melt conditions and ii) the different melt-induced snow and ice properties
in the Southern Hemisphere comprising a larger fraction of coarse-grained snow due to pro-longed melt-freeze cycles and a
generally drier snow surface, at least for the high-concentration parts of the sea-ice cover.

On the one hand, the negative biases (Figure 9, Table 13) agree well with results of earlier comparisons between

Southern Hemisphere summer PMW SIC estimates and ship-based observations of the sea-ice cover (e.g. Worby and Comiso,
2004; Ozsoy-Cicek et al., 2009). These studies hypothesize that under-estimation of the actual sea-ice concentration in PMW
SIC products is primarily caused by wet, flooded sea ice exhibiting a similar surface emissivity as open water and hence
looking like open water in PMW imagery. On the other hand, an unknown fraction of these negative biases could be caused
by our Landsat SIC estimates being biased high because of the reasons laid out in Subsection 2.2.4 and the respective
supplementary material.

**Table 13.** Summary of statistical results obtained for 45 melt-conditions cases in the Southern Hemisphere. See caption of
Table 5 for an explanation of the parameters given. Numbers added behind the "/" for SICCI-2 products and OSI-450 denote
results obtained using non-truncated SIC.

| LS8, SH 2013-15 | SICCI-12 | SICCI-25 | SICCI-50 | OSI-450 | CBT-SSMI | NOAA-CDR | CBT-AMSR2 | NT1-SSMI | ASI-SSMI | NT2-AMSR2 |
|---|---|---|---|---|---|---|---|---|---|---|
| Diff | -5.0 / -4.3 | -5.8 / -5.5 | -8.1 / -7.8 | -4.9 / -4.6 | **-0.4** | ***-0.6*** | -2.8 | -8.7 | -7.8 | 5.1 |
| DiffSDEV | **13.7** / **14.1** | ***13.9*** / 14.1 | 17.1 / 17.2 | 14.8 / ***14.9*** | 15.6 | 15.6 | 15.4 | 16.4 | 18.6 | 15.9 |
| Slope | 0.888/0.903 | ***0.951/0.958*** | 0.983/0.991 | 0.750/0.754 | 0.772 | 0.794 | 0.895 | 0.791 | 0.859 | 0.824 |
| Intercept | 4.0 / ***3.5*** | -1.8 / -2.1 | -6.7 / -7.1 | 14.1 / 15.4 | 18.0 | 16.0 | 5.8 | 8.2 | ***3.6*** | 19.4 |
| R² | **0.79** / **0.78** | ***0.78*** / **0.78** | 0.69 / 0.69 | 0.71 / 0.71 | 0.69 | 0.69 | ***0.72*** | 0.67 | 0.65 | 0.69 |
| N | 10214 | 2915 | 916 | 2915 | 2899 | 2899 | 2955 | 2929 | 10129 | 2955 |




## 5 Discussion

### 5.1 A note on grid resolutions

SICCI-25km and SICCI-50km SIC have a grid resolution close to the actual algorithm resolution largely determined by the native resolution of the lowest-frequency channel used (see field-of-view dimensions in Table 1). This is not the case for, e.g. CBT-SSMI or OSI-450. Actually, we find a relatively poor performance of OSI-450 in comparison to SICCI-25km (see Tables 5 to 7) – albeit the retrieval algorithm is exactly the same. We hypothesize that the coarser native resolution of the satellite data used for OSI-450 provides one of the main explanations for this observation. SICCI-25km uses AMSR-E and AMSR2 brightness temperatures observed at spatial resolutions (footprint sizes) between 14 km x 8 km (AMSR2: 12 km x 7 km) and 27 km x 16 km (AMSR2: 22 km x 14 km) (see Table 1). In contrast, OSI-450 uses SSM/I and SSMIS brightness temperatures observed at footprint sizes between 37 km x 28 km and 69 km x 43 km. In addition, the relevant channels are sampled spatially every 10 km for AMSR-E / AMSR2 and every 25 km for SSM/I / SSMIS. Therefore, spatial brightness temperature variations caused, e.g., by variations in the open water fraction, can be identified at a finer spatial scale by AMSR-E / AMSR2 than by SSM/I / SSMIS at the same frequency. The grid spacing at which OSI-450 and other SIC products relying on SSMI(S) 19 / 37 GHz channels are provided is not the actual resolution of the estimated SIC. Surface information is smeared in the SSMI(S) data much more. A similar observation applies to CBT-SSMI and CBT-AMSR. The latter provides SIC at a grid resolution, which is closer to the algorithm resolution than that of CBT-SSMI; consequently, CBT-AMSR SIC agree closer to Landsat SIC than CBT-SSMI SIC (see Tables 5, 6, and 7 and compare panels e) and g) in Fig. 2, 3 and 4). We are confident that, besides the differences between the algorithms themselves, a substantial fraction of the observed difference in the agreement with Landsat SIC is caused by the spatial representation of the true sea-ice concentration, which differs due to the above-mentioned differences in satellite data used as input.

Our results confirm the stated impact of the native spatial resolution on potential biases between PMW SIC and Landsat SIC very well. For instance, out of the ten products, ASI-SSMI and SICCI-12km both incorporating high-frequency, fine spatial resolution imagery channels provide the 3[rd] and 4[th] best linear fits in the Northern Hemisphere (Tables 5, 6) and the 3[rd] and 5[th] best linear fits in the Southern Hemisphere. When compared to the other SICCI-2 products, SICCI-12km has considerably better results in the Southern than the Northern Hemisphere; SICCI-12km actually performs best out of the four SICCI-2 / OSI-450 products (Table 7, Table S04 in the supplementary material). Our Landsat data set of the Southern Hemisphere contains more cases of ice regimes (see Section 4) with variable open water fractions such as "heterogeneous ice", "leads / openings", "freeze-up", and "ice edge" than the one of the Northern Hemisphere (see Table S01 in the supplementary material). Because a SIC product at finer spatial resolution is capable to depict such variable open water fractions better and to observe the full SIC range more adequately it seems reasonable to have a better linear agreement between Landsat SIC and, e.g., SICCI-12km SIC in the Southern than the Northern Hemisphere (compare Figs. 3 and 4 with respect to low SIC).

However, the other PMW SIC product with 12.5 km grid resolution, ASI-SSMI, does not show better results in the Southern than the Northern Hemisphere when compared to, e.g. NT1-SSMI or SICCI-2 products. ASI-SSMI utilizes near-90 GHz brightness temperatures only while SICCI-12km combines near-90 GHz with 19 GHz brightness temperatures. Atmospheric effects known to cause biases in near-90 GHz PMW SIC products (Kern, 2004; Ivanova et al., 2015) might therefore have less impact on SICCI-12km than ASI-SSMI SIC. In addition, all SICCI-2 products utilize brightness temperatures corrected for atmospheric effects using radiative transfer modelling while ASI-SSMI utilizes uncorrected brightness temperatures. The fact that most of our Landsat scenes in the Southern Hemisphere represent atmospheric conditions during summer melt and hence at a comparably higher water vapor load than in the Northern Hemisphere fits into this picture. While the atmospheric effects are efficiently mitigated for SICCI-12km in both hemispheres these are larger for ASI-SSMI in the Southern than the Northern Hemisphere.





### 5.2 Hemispheric differences versus Landsat SIC bias

At this point, we look at the difference between the SIC differences obtained in the Northern Hemisphere and the Southern Hemisphere from a different perspective. Ice conditions represented by our Landsat SIC data set comprise more cases with melt conditions and at the ice edge in the Southern Hemisphere (see Table S01 in the supplementary material). These conditions are likely particularly subject to the positive bias in Landsat SIC due to mixed pixels described in Subsection 2.2.4 (see also the respective supplementary material). Therefore, we can expect that the positive SIC difference is, on average, larger in the Southern than the Northern Hemisphere. We compare the differences listed in Tables 5, 6 and 7 and find the following. OSI-450, SICCI-12km, and SICCI-25km exhibit small changes in the difference PMW SIC minus Landsat SIC between +0.8 % and -0.8 %. NT2-AMSR reveals a positive change of +2.8 %. Both CBT products, NOAA-CDR, NT1-SSMI, ASI-SSMI, and SICCI-50km show a negative change by between -2.2 % and -3.2 %. This change of ~ 3 % in the SIC difference between the results of the Northern and the Southern Hemisphere is of the correct sign and of an order of magnitude we deem a realistic estimate of the difference in the mentioned positive Landsat SIC bias between the hemispheres. What does this mean? For example, for a PMW grid cell covered by an actual SIC of 95 %, due to the positive bias Landsat SIC might be 97 % in the Northern Hemisphere and 100 % in the Southern Hemisphere. A PMW SIC algorithm tuned equally well for the ice conditions in the respective hemisphere would provide 95 % in both hemispheres. Compared to Landsat SIC this results in a negative difference of -2 % in the Northern Hemisphere and of -5 % in the Southern Hemisphere, i.e. the difference becomes even more negative. In contrast, the difference NT2-AMSR SIC minus Landsat SIC becomes even more positive, increasing from +0.6 % in the Northern Hemisphere to +3.4 % in the Southern Hemisphere. When only considering the melt-condition cases the overall difference increases from +1.7% to +5.1% (Tables 12, 13). Without further independent evaluation data to better assess the accuracy of our Landsat SIC data we cannot draw a quantitative conclusion here. However, the increase in the positive value of the difference PMW SIC minus Landsat SIC between the Northern and the Southern Hemisphere observed for NT2-AMSR is opposite to our well-motivated suggestion that Landsat SIC values are biased higher in the Southern than the Northern Hemisphere.

### 5.3 A note on the effect of filters

In this subsection, we comment on the observation that in the scatterplots of the Northern Hemisphere (Figs. 2 and 3) particularly the SICCI-2 products but also OSI-450, CBT-AMSR and NT2-AMSR exhibit a relatively large number of cases with PMW SIC = 0 % and Landsat SIC > 0 %. In addition, we find an unexpected high number of comparably low PMW SIC values (< ~ 50 %) at Landsat SIC > ~ 70 %, especially for SICCI-50km (Fig. 2c, Fig. 3c). In the scatterplots of all products in the Southern Hemisphere (Fig. 4) we observe a large number of cases with PMW SIC = 0 % and Landsat SIC > 0 %.

We hypothesize this observation is linked to the various filters applied. Examples of such filters are the weather or open water filter (OWF) and the land spill-over filter (LSO). The OWF reduces the number of erroneous SIC values resulting from unaccounted atmospheric influence, for example high cloud liquid water contents. OWF is effective along the ice edge and the adjacent open water. One common realization of the OWF is to set PMW SIC values to 0 % SIC once brightness temperature gradient ratios sensitive to the atmospheric influence exceed a certain threshold (e.g. Wensnahan et al., 1993; Spreen et al., 2008; Lavergne et al., 2019). Such filters might cut off true SIC values (Andersen et al., 2006). The SICCI-2 and OSI-450 algorithm employs a modified version of such an OWF (Lavergne et al., 2019; Kern et al., 2019). The LSO reduces the number of erroneous SIC values along coastlines resulting from unaccounted spillover of the (higher) land surface brightness temperature into the (lower) open water brightness temperature. The LSO is particular effective during summer. It has also an influence during the freezing season for situations where the coastline is only fringed by a quite narrow sea ice cover, as is the case during fall freeze-up in the Hudson Bay or along the Siberian coast or during winter / spring along the coast of Greenland facing the Irminger Sea. One realization of the LSO is a statistical approach, where the SIC of grid cells adjacent to the coast is corrected, i.e. set to 0 % or interpolated to a more adequate value, based on SIC values within a certain neighborhood (e.g.



Cavalieri et al., 1999). The SICCI-2 and OSI-450 algorithm employs a novel attempt. Here the method of Maass and Kaleschke
(2010) is used to correct for the land spillover already at the level of the brightness temperatures input to the SIC retrieval; the
"classical" LSO filtering of Cavalieri et al. (1999) is still included, though (Lavergne et al., 2019). Note: the OWF sets PMW
SIC to zero; the LSO reduces the PMW SIC to lower values but not necessarily to zero.

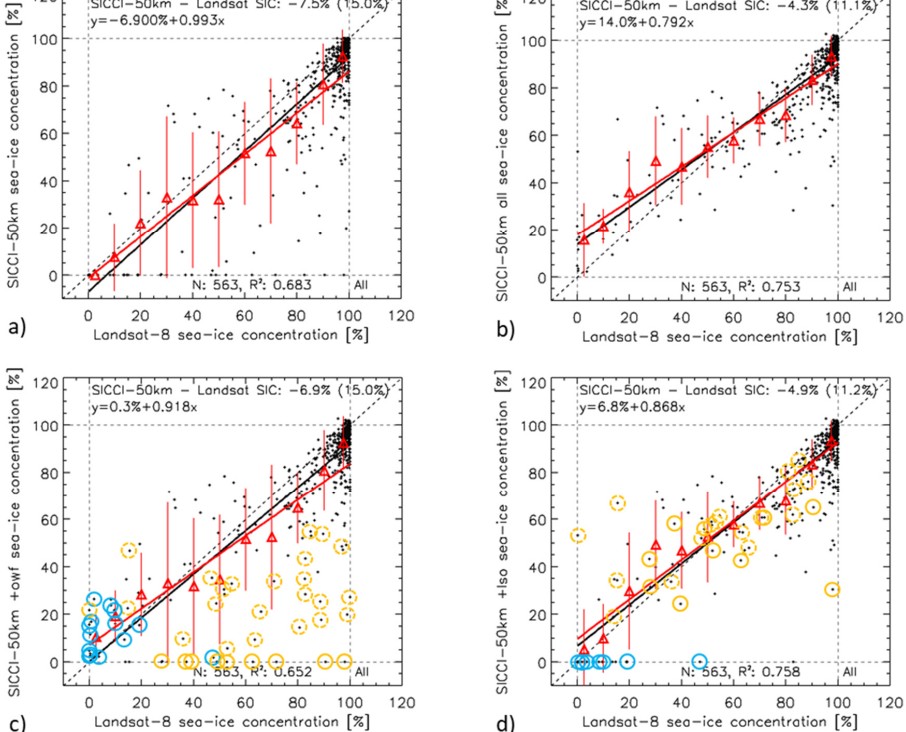

**Figure 10.** Scatterplots of SICCI-50km SIC (y-axis) versus Landsat SIC (x-axis) for ice regime "leads/openings" in the
Southern Hemisphere in years 2013-2015. Black dots are individual data pairs, the black solid line is the linear regression, and
the black dashed line is the identity line. Red triangles denote the mean PMW SIC computed for Landsat SIC ranges 0%-5%,
5%-15%, 15%-25%, … , 85%-95%, 95%-100%, the red bars one standard deviation of these mean values; the red line is the
respective linear regression line. The overall difference PMW SIC minus Landsat SIC, its standard deviation, and the equation
for the linear regression using the individual data pairs is given at the top, the number N of data pairs and the squared linear
correlation coefficient at the bottom of each panel. Panel a) Fully truncated SIC, all filters applied; panel b) fully non-truncated
SIC, no filters applied; panel c) truncated / non-truncated SIC, GT100 and OWF applied; panel d) truncated / non-truncated
SIC, GT100 and LSO applied. Blue circles mark SICCI-50km SIC values set to 0 % by the OWF; orange circles mark SICCI-
50km SIC values set changed by the LSO (solid circle: SIC set to 0 %, broken circle: SIC reduced).

The SICCI-2 and OSI-450 products offer the full SIC distribution around 0 % and around 100 % SIC and the

opportunity to reverse-engineer the effect of flags, i.e. switch the effect of certain flags on or off. Therefore, we are able to
investigate the impact of the OWF and the LSO on our comparison results, an investigation not possible for the six other
products. We choose the ice regime "leads/openings" in the Southern Hemisphere in years 2013-2015 and look, as an example
for such an investigation, at the impact of the two above-mentioned filters on SICCI-50km SIC in comparison to Landsat SIC
(Fig. 10). Note that we switch off these flags together with the near-100 % SIC flag to work with a more realistic SIC





distribution at the high-concentration end. The first finding is that there is not even one PMW SIC = 0 % in the fully non-
truncated, i.e. no filters applied, SIC scatterplot (Fig. 10b) – in contrast to the fully truncated SIC (Fig. 10a). Accordingly, the
statistical parameters differ considerably. For instance, the overall SIC difference reduces in magnitude from 7.5 % for the
fully truncated version of SICCI-50km to 4.3 % for fully non-truncated version; the standard deviation of the difference reduces
from 15.0 % to 11.1 %.

If we switch off the OWF, i.e. include the originally retrieved SIC values for those grid cells where the OWF is applied,

we get a number of SIC data pairs concentrated between Landsat SIC: 0 – 20 % and SICCI-50km: 0 – 30 % that can be clearly
associated with the OWF (compare Fig. 10 panel c) with panels a) and d). Statistical parameters change little. For instance, the
magnitude of the difference decreases by 0.5 % while the standard deviation stays the same. There is still a comparably large
number of cases with SICCI-50km SIC = 0 % or at least relatively low: < 30 %, concomitant with Landsat SIC > 50 %. If we
instead switch off the LSO, i.e. include the originally retrieved SIC values for those grid cells where the LSO is applied, we
find that almost all of the above-mentioned cases of low or equal-to-0 % SICCI-50km SIC can be traced back to substantially
higher SIC values (Fig. 10d). Statistical parameters change considerably. For instance, the magnitude of the difference changes
from 7.5 % (see above) to 4.9 % if keeping only the LSO filtered grid cells; the standard deviation of the difference reduces
from 15.0 % (see above) to 11.2 %. This reduction in the spread of values around the identity line is also evident very well in
the respective scatterplots (Fig. 10): the standard deviation of the Landsat SIC 10 % bin average SICCI-50km SIC (red vertical
bars) is much smaller in panel d) than panel a).

We observe a similar tendency for all other ice regimes where the LSO is applied, e.g. "freeze-up" or "melt conditions",

in the Southern and in the Northern Hemisphere and for SICCI-25km and SICCI-12km as well (see Tables S04 and S05 in the
supplementary material). We note, however, that there are far fewer SIC data pairs subject to LSO filtering for OSI-450; hence
the effect of switching on or off the LSO is comparably small. We hypothesize that this could be explained with the different
native resolution of the satellite data used, the different sampling, and the different grid cell size and spacing. However, testing
this hypothesis is beyond the scope of this paper. For the SICCI-2 SIC products, we can confirm the hypothesis that the
comparably large number of PMW SIC = 0 % or < ~30 % across basically the entire SIC range (see Figs. 2, 3, and 4, panels
a) to c) can be explained with the application of an LSO resulting in an elevated number of cases with PMW SIC smaller than
Landsat SIC. This provides a viable explanation for unexpectedly large SIC differences observed for SICCI-50km along
coastlines, of particularly Greenland or the Eastern Antarctic, reported in Kern et al. (2019, their Fig. 8 c) and Fig. 11 c).
Whether this is due to the land spillover correction at the brightness temperature level (Maass and Kaleschke, 2010) or the
statistical filtering (Cavalieri et al., 1999) remains to be investigated. We clearly see it as an advantage that for SICC-2 and
OSI-450 products we can switch off filters and in a reverse-engineering way investigate the impact these filters have on PMW
SIC. This appears not to be possible for the remaining six PMW SIC products. Application of the LSO can produce an
artificially large number of SIC values near at or 0 % that agree less well with the Landsat SIC than the originally retrieved
SIC values – as we demonstrate for the SICCI-2 and OSI-450 products. As a consequence, results of an evaluation including
a considerable number of near-coastal grid cells need to be interpreted carefully. The number of artificially low SIC values
resulting from the LSO for the other six PMW SIC products is unknown as is their impact on the evaluation results shown in
this paper.
**6    Summary and Conclusions**

In this paper, we present results of an evaluation of ten passive microwave (PMW) SIC products against SIC estimates

derived from a total of more than 300 clear-sky Landsat visible images acquired in the Northern Hemisphere during mostly
late winter / spring (March through May) and in the Southern Hemisphere during spring / summer / early fall (October through
March). We estimate Landsat SIC at the grid resolution of the PMW SIC products using results of supervised classification of



Landsat broadband albedo maps into ice and water at 30 m pixel resolution. The comparison between PMW and Landsat SIC
is carried out based on all valid collocated SIC map pairs but also based on subsets of these pairs defining certain ice regimes.
These ice regimes are "high concentration", "freeze-up", "ice edge", "leads/openings", "heterogeneous ice", and "melt
conditions".

Our comparison uses parameters such as the mean difference between PMW and Landsat SIC and its standard deviation,

the median difference, and parameters describing the linear agreement: slope and intercept of a linear regression and the linear
regression coefficient. We summarize these parameters in Figures 11 and 12 and come up with the following conclusions.

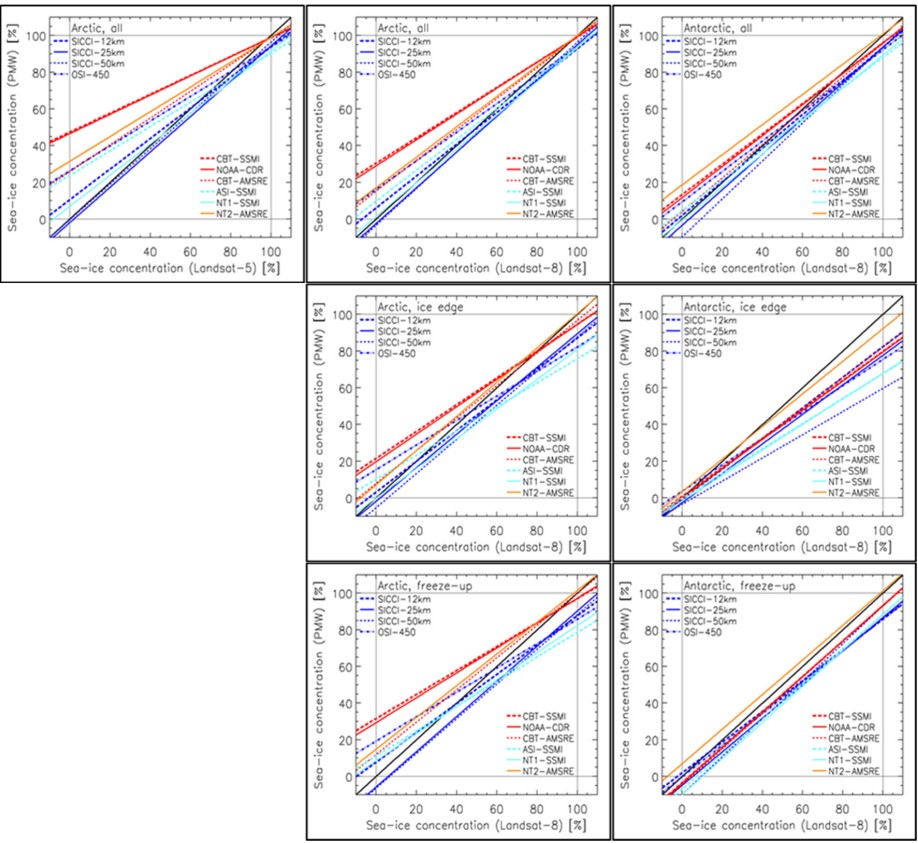


**Figure 11.** Summary of all linear regression lines obtained for the comparison between Landsat SIC and PMW SIC for all ice
regimes – except high-concentration ice. Columns denote, from left to right, Landsat-5 Arctic (i.e. first-year ice), Landsat-8
Arctic (i.e. mixed first-year / multiyear ice and multiyear ice), and Landsat-8 Antarctic. Ice regimes are sorted per row from
top to bottom: "all" cases, "ice edge", and "freeze-up". Different colors and line styles denote different products as indicated.
The black solid line denotes the identity line.

788



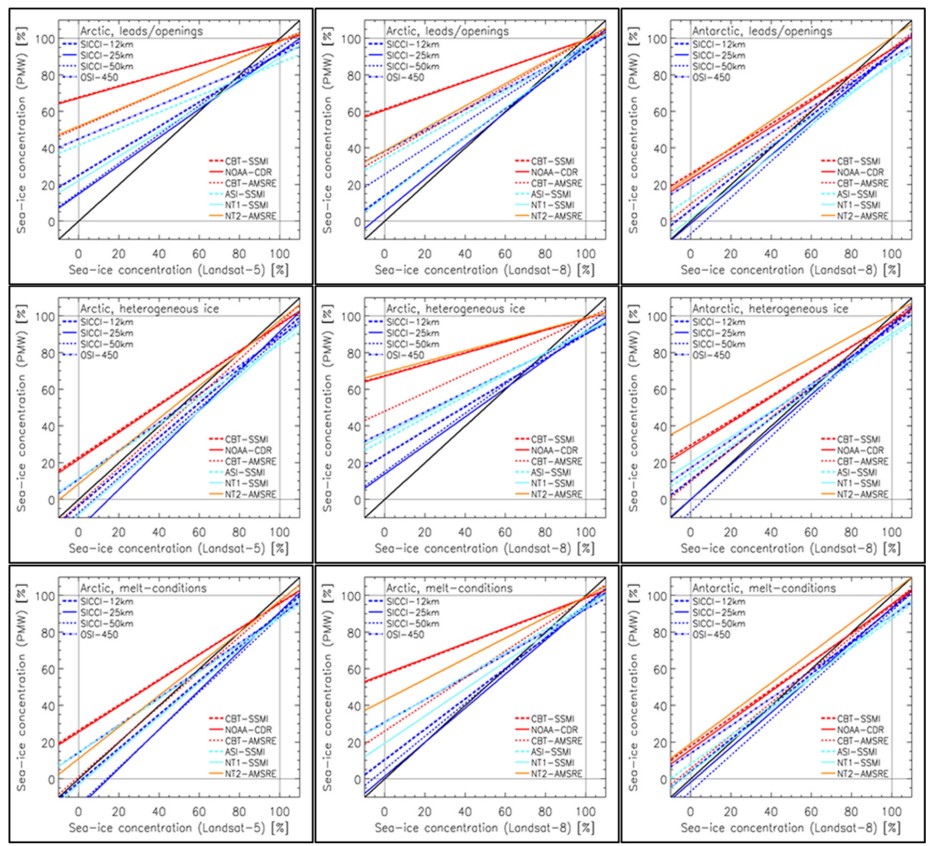

**Figure 11.** continued for ice regimes "leads and coastal openings", "heterogeneous ice", "melt-conditions".

- It is important to take an integrated view of these parameters because, for instance, a small overall bias is not necessarily associated with a good linear agreement across the entire SIC range and a perfect linear agreement with a slope close to unity and a high correlation could be associated with a large overall bias.

- It is also important to take into account the expected influences of, e.g. melt conditions (section 4.3), fraction of new/thin ice (section 4.1) as well as sub-pixel size ocean-ice mixture (Section 2.2.4) on both PMW SIC and Landsat SIC.

- SICCI-25km and SICCI-50km SIC offer overall the best linear agreement to Landsat SIC as demonstrated in Fig. 11 and Fig. 12, right column. This is illustrated as well by mean and median PMW SIC values computed for Landsat SIC bins aligned very well along the identity line (Figs. 2 to 4), with exceptions being explainable by filters applied in the products (see Section 5.3). The magnitude of the difference PMW SIC minus Landsat SIC is, however, larger than for the two CBT-products and NOAA-CDR, almost without exception (Fig. 12, left column).

- The two CBT products, NOAA-CDR and NT2-AMSR offer the smallest overall magnitude of the difference PMW SIC minus Landsat SIC (Fig. 12, left column). Except for CBT-AMSR2 in the Southern Hemisphere, mean and median PMW SIC values align less well along the identity line than for SICCI-25km and SICCI-50km in Figs 2 to 4. The linear agreement is considerably worse than for SICCI-25km and SICCI-50km (Fig. 11, Fig. 12, right column).

- NT2-AMSR is the only product over-estimating Landsat SIC in the Southern Hemisphere – overall but also for almost all ice regimes. This is problematic in view of the potential positive bias of Landsat SIC for ice conditions with an elevated number of mixed ocean-ice Landsat pixels (see Subsection 2.2.4), e.g. ice regimes "melt conditions", "ice edge" and "freeze-up".

810

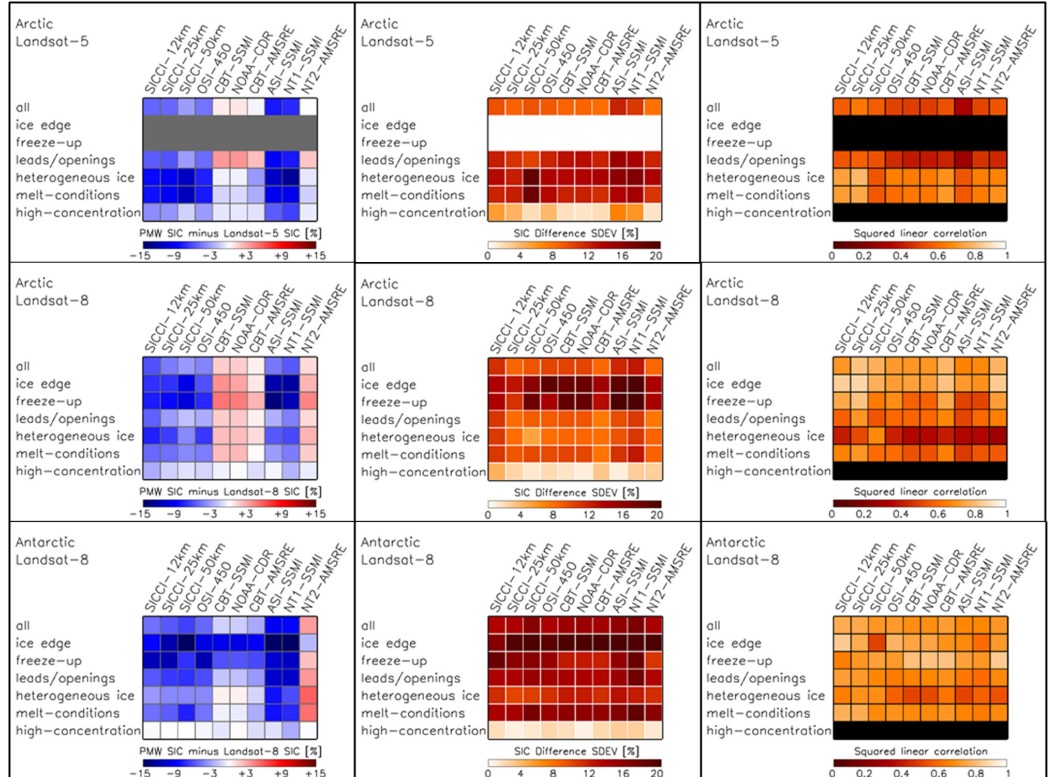

811

**Figure 12.** Illustration of the statistical parameters of the comparison between Landsat SIC and PMW SIC for all ice regimes.
Rows denote, from top to bottom, first-year ice Arctic (Landsat-5), mixed first-year / multiyear ice and multiyear ice Arctic
(Landsat-8), and all ice Antarctic (Landsat-8). Columns denote, from left to right, accuracy (difference PMW SIC minus
Landsat SIC), precision (standard deviation of the SIC difference), and squared linear correlation coefficient. The uni-colored
rows denote cases left out, either because these ice regimes are not populated (topmost row of panels) or because the retrieval
of parameters did not make sense (Squared linear correlation for ice regime "high concentration").

All products provide SIC data truncated to the range 0 % to 100 % albeit all algorithms but NT2-AMSR use a SIC
retrieval procedure which in principle provides a full SIC distribution around the end-members 0 % and 100 %. Only the
SICCI-2 products and OSI-450 allow consideration of the full SIC distribution. While our main results are derived with the
truncated SIC distribution, we demonstrate that, without exception, using the full SIC distribution reduces the mean difference
and enhances the quality of the linear agreement between PMW SIC and Landsat SIC which is already superior for SICCI-
25km and SICCI-50km. It is important to consider when comparing the results obtained with the ten products against each
other in order to avoid misinterpretation. While we obtain smallest SIC differences for the two CBT products, NOAA-CDR
and NT2-AMSRE/2, this is likely to change using the full SIC distribution. This applies in particular to ice regimes "high-
concentration" (section 4.2) and "melt conditions", but also to the full set of SIC data pairs (denoted "all" in Fig. 12). The
impact this difference in the comprehensiveness of the SIC products has on our evaluation results prevents us from making a
ranking between the SIC products.

This paper is limited to clear-sky visible imagery. It is hence impossible to evaluate the performance of the SIC products
under the full set of possible weather conditions with an influence on the SIC retrieval, i.e. surface wind speed and atmospheric
water vapor and cloud liquid water content. We note that our results likely cover a certain range of surface wind speeds and



atmospheric water vapor contents which we, however, did not quantify, e.g. by means of atmospheric reanalysis data, to stay

focused. Obviously, this would be an issue worth pursuing in a forthcoming study for which SIC estimates based on SAR data

have to be used. These might allow to assess PMW SIC quality also under higher loads of atmospheric water vapor content

and, more importantly, clouds. Such a study could then focus in particular on an improved accuracy assessment of the PMW

SIC in the marginal ice zone and along the ice edge. In such regions, our approach to derive Landsat SIC likely results in the

highest positive biases – between a few to in the worst case 20 % for single PMW grid cells – due to mixed ocean-ice Landsat

pixels classified as ice. Such a study would also be an excellent opportunity to evaluate the weather filters currently employed

in the SIC products. In order to have a meaningful sample, such a study would require an equally large number of SAR images

interpreted into well-evaluated SIC estimates for a number of years covering both hemispheres as is used in this paper. This

calls for continued development of reliable and consistent SIC estimates from SAR and, thorough evaluation of SAR SIC

products in both hemispheres.

*Data availability.* All sea-ice concentration products except SICCI-12km are publicly available from the sources provided in the reference list or in Kern et al. (2019). The SICCI-12km product is available upon request from T. Lavergne. The classified Landsat images are available from https://doi.org/10.25592/uhhfdm.9181 (last access: July 9 2021).

*Author contributions.* SK wrote the manuscript. TL, LTP and RT contributed to the concept and work presented in the paper and also assisted in the writing. SK performed the data analysis together with LB, MM, and LZ. SK conducted the inter-comparison with contributions in the interpretation of the results from TL, LTP and RT.

*Competing interests.* The authors declare that they have no conflict of interest.

*Acknowledgements.* The work presented here was funded by EUMETSAT (through the 3rd Continuous Developments and Operation Phase of OSI SAF) and ESA (through the Climate Change Initiative Sea_Ice_cci project), and the German Research Foundation (DFG) Excellence Initiative CLISAP under Grant EXC 177/2. The publication contributes to the Cluster of Excellence 'CLICCS – Climate, Climatic Change, and Society' and to the Center for Earth System Research and Sustainability (CEN) of the University of Hamburg.

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
