# Peer review of "Satellite Passive Microwave Sea-Ice Concentration Data Set Inter- 2 comparison using Landsat data"

_The Cryosphere, 2021_

## Author Comment (AC1)

*Title: Satellite Passive Microwave Sea-Ice Concentration Data Set Inter-comparison using Landsat data*

*Author(s): Stefan Kern et al.*

*MS No.: tc-2021-258*

*MS type: Research article*

*---*

*General Comments*

*The authors compare sea ice concentration from 10 global passive microwave products with that estimated from visible/near-infrared Landsat images. This highly detailed analysis is an important piece of work: the passive microwave record constitutes the longest existing record of global sea ice cover and it is crucial that users of these data understand when and where we have good confidence in particular products. One particularly useful aspect of this work is the comparison with the range of different ice regimes and an assessment of confidence in this comparison. This work highlights the need to assess satellite products under a range of different ice conditions and to consider concomitant processes that influence the sea ice concentration.*

*The authors provide good context by highlighting some of the limitations of the previous work on which this study builds and highlighting the reasons why a comparison of products such as this one is needed. They also provide the background and support needed to justify the datasets they are using to make the comparison.*

*This is a robust piece of work. The authors have given thorough consideration to any bias their method introduces to the estimation of sea ice concentration in the Landsat images and have made a rigorous comparison for different ice types. The case studies provide …*

*One thing I am not sure about is the accessibility to a wide readership. It is (necessarily) a dense read packed with technical details but as a reader, it is quite hard to keep track of all this detail. However, this is an important and well-conducted contribution to the literature and I recommend this manuscript for publication. I have noted some minor points below.*

Thank you for your honest and positive opinion about the manuscript. We are aware that it is (again) quite long as the two previous publications of the same kind. This is an opinion also shared by the second reviewer. To our opinion, however, it is the degree of detail which makes this publication of interest to those readers that are into the topic, because they might find the missing piece of information they have been looking for without success in other papers about a similar topic. In order to enhance the readability for readers with a general interest only, we tried to be as succinct as possible in abstract and summary.

*Line 45-46: I am not sure what you mean by "convergent high-concentration ice situations".*

Changed to "… pairs revealing convergent ice motion and therefore a high probability to encounter high, near-100% SIC."

*Fig 11: it is quite hard to read the text and there is no accompanying table. Perhaps you could move the legend into the white space below the Landsat 5 figure and increase the size of the text?*

As Fig. 11 has a second part without space below the Landsat 5 figure, we reshuffled the legends such that we could also increase the font size.

*Line 43: suggest changing "are" to "a"*

Thank you, we corrected this typo.

*Line 44: you might consider changing the first instance of "used" to "assessed".*

We don't think "assessed" fits here and instead replaced "used" by "utilized".

*Line 85: you could just quote the number here to aid the reader, e.g. "an order of magnitude smaller than the 300 scenes used in this study".*

Thank you. This is a good idea, which we realized as suggested.

*Line 101: suggest changing "satisfying" to "satisfactory".*

Changed as suggested.

*Line 107: suggest adding "that are" in between "polygons" and "highly".*

Changed as suggested.

*Line 121-123: there is something not quite right with the grammar here.*

We changed this sentence into: "The same applies to the fact that four of the products (SICCI-12km, SICCI-25km, SICCI-50km, and OSI-450) allow us to take into account the full SIC distribution at 0 % and 100 %. Such a distribution is the natural result of the SIC retrieval method used in all SIC products considered – except NT2-AMSR."

*Line 127: suggest changing "of" to "from".*

Changed as suggested.

*Line 130: suggest changing the second instance of "of" to "from".*

Changed as suggested.

*Line 153-154: suggest changing both instances of "is" to "are".*

Thank you, we corrected these typos.

*Line 296: suggest changing "quantifying" to "to quantify".*

Changed as suggested.

*Line 308: do you mean "cases" instead of "cased"?*

Certainly we do. Typo is corrected.

*Line 599: suggest "also is" instead of "is also".*

Changed as suggested.

*Line 642: suggest deleting ", which is".*

Deleted as suggested.

---

## Author Comment (AC2)

*Review of "Satellite passive microwave sea-ice concentration data set inter-comparison using Landsat data", by Kern et al.*

*Summary*

*This paper presents a comprehensive comparison of ten different passive microwave sea ice concentration products with Landsat visible imagery. Comparisons are made in both hemispheres and over a wide range of spring through autumn conditions. The results indicate varying performance of the algorithms, with the SICCI providing the best linear agreement with Landsat across different concentrations. The CBT, NOAA-CDR, and NT-2 AMSRE have the smallest overall biases, but this may relate to truncation of values at 0% and 100%; the NT2 substantially overestimates concentration in the Antarctic.*

*General Comment*

*This is a very in-depth and comprehensive paper and further adds to the excellent passive microwave sea ice comparisons studies over the past couple of years (Kern et al., 2019; Kern et al., 2020). The study is thorough and laid out well. My only main criticism is that it is a very bulky manuscript and difficult to take all of it in. I note in the comments below that only three of the six case studies are presented in the main manuscript, with the other three relegated to the supplement. This is fine, but it seems to make a somewhat arbitrary (maybe there is a rationale?) split between what is in the main paper and what is in the supplement. And the supplement itself is quite extensive – 18 pages, albeit a lot of that is figures and tables. I wonder if there might be value in splitting the paper into two parts – one paper with the main hemispheric comparisons and then a second paper examining the case studies? I would leave this up to the authors and the editor, but it may be something to consider.*

Thanks a lot for this useful and critical remark. We read the manuscript thoroughly with the aim to investigate where we could i) shorten the supplement and ii) provide reasonable argumentation about why we include the three ice regimes chosen into the main manuscript while keeping the others in the supplementary material.

Regarding i) our opinion is that we can merge Table S04 with Table 8, hence include it in the main manuscript. We also think that Figure S03 is not required and can be omitted.

Regarding ii) we stated in the first paragraph of Section 4 that the three ice regimes enclosed in the main manuscript are actually those from which we know that SIC biases are likely to occur. We will stress this piece of information better in the final manuscript as this is our main motivation to show examples of these three ice regimes in the main manuscript.

One could think of omitting ice regimes "ice edge" and "leads & openings" in general. On the one hand, with the clear-sky Landsat SIC data set at hand, we can only provide limited information about the

accuracy of the PMW SIC for these two regimes likely impacted by clouds more than the other regimes. On the other hand, these regimes are important to illustrate that the filters (open water filter, OWF; land spill-over filter, LSO) used by the ten products can bias the results of an inter-comparison of the kind carried out like we discuss in Section 5.3. Note that ice regime "leads & openings" includes many cases close to the coast where LSO has an impact. In addition, we find that same algorithms using data at different native resolutions provide the expected improvement in the agreement between PMW and Landsat SIC (SICCI-25km vs. OSI-450 and CBT-AMSR vs. CBT-SSMI) for regime "ice edge".

Ice regime "heterogeneous ice" is the least well-defined regime and is a regime for which we do not provide an additional motivation about why this is important to include in the overall assessment. Therefore, this would be the ice regime for which it would be most logical to exclude it entirely from the analyses shown (both in the supplementary material AND in Figs. 11 and 12). However, this is the ice regime where one would expect to see a dependency of the results as a function of native and grid resolutions – similar to ice regime "ice edge". We find evidence of this expectation in Fig. 12 but it is not unique and depends on the hemisphere and ice type we look at. To our opinion, this points to future work. We will stress this issue better in the revised manuscript.

Because of these reasons we think it is reasonable to keep sample figures for ice regimes "ice edge", "leads & openings" and "heterogeneous ice" in the supplementary material because these provide valuable examples of how the ten SIC products map sea-ice conditions for these regimes of which we have quite a large number of cases (see Table S01).

What prevents us from following a two-paper concept is that we are at a stage where more, finer resolution SIC products are emerging / gaining length and where established SIC products like the NOAA/NSIDC CDR already came up with new version. We are convinced that it would be more valuable to elaborate on the results presented in this paper in forthcoming studies focusing on particular elements and open questions articulated in this paper and apply the knowledge gained to a novel set of SIC products and a novel set of evaluation data.

*I have some fairly minor comments below that should be addressed by the authors. I recommend acceptance after minor revisions.*

*Specific Comments (by line number):*

*127-128: In Table 1, the CBT and NT2 AMSR-E/2 products used are at 25 km resolution. But there is also a 12.5 km resolution product. Is there a reason to use the lower resolution product over the 12.5 km product? Higher resolution will pick more detail and generally be more accurate. I can see for simplicity picking only one or the other as the differences shouldn't be too large, but the 12.5 km product would make more sense to me and would have been a 2nd12.5 km product along with the SICCI-12km and ASI. Perhaps you wanted to be consistent with the SSMIS 25 km products?*

Thank you. This is an absolutely logical question. We have several points that – as we hope – could assist you in a better understanding of our motivation.

We have three flavors of the SICCI SIC products, one at 12.5 km, one at 25 km and one at 50 km grid resolution. The one at 25 km resolution utilizes AMSR-E / AMSR2 data of channels similar to the AMSR-E/2 versions of CBT and NT2. We chose a grid resolution of 25 km for SICCI-25km because: i) the actual resolution of the product – determined by the footprint size of the channels used – is actually closer to 25 km than to 12.5 km; ii) the noise in the retrieved SIC is lower when there is a closer match between the spatial and the grid resolution. A low noise is one of the primary objectives of a CDR. Therefore, in order to compare a similar set of products in terms of the noise inherent in the SIC products originating from differences between spatial (footprint, sampling) and grid resolution we chose those AMSR-E/2 products also at 25 km grid resolution. We believe that it is more consistent to do so.

Furthermore, the majority of the products compared here come at 25 km grid resolution. We found it particularly useful to compare the OSI-450 SIC CDR and the NOAA/NSIDC SIC CDR (and its components) that are based on SSM/IS data with products at the same grid resolution. We find it interesting to see how at the same grid resolution of 25 km the results based on AMSR-E/2 data tend to be superior over those based on SSM/IS data (even though this has been shown for CBT and NT2 in previous work of other authors).

Another, while considerably weaker argument is that we also wanted to be consistent to the two previous papers where we compared the same set of algorithms and products.

In the future, in case ESA-CCI SIC CDR product would be released at finer grid resolution, we would incorporate the 12.5 km products you mentioned and finer resolved versions of the ASI-algorithm (6.25 km or even 3.125 km).

*138: In Table 1, the references seem to ATBDs or journal papers about the products. However, one should also reference the products themselves where available. I do see such references in the Reference list – e.g., Meier et al., 2017 for the NOAA CDR. However they are not listed in Table 1 or within the manuscript text (as far as I can tell) – e.g., in Table 1, for the NOAA CDR, the references provided are for the ATBD and a journal article. I would suggest adding the actual product citation in the far right column, again if available.*

Thank you; we added the respective references also in Table 1 were appropriate.

*Also, I will note that the NOAA CDR used here is apparently Version 3. This is fine and there is no need to change. But I will note that there is now a Version 4 published that has some notable differences from Version 3, though nothing that I think would substantially affect your results. Nonetheless this highlights the need to cite the specific dataset, including the version, where possible so that there is clarity in what data is being used.*

Yes, absolutely. There is a now a version 4 published and we are eager to include this new version into forthcoming inter-comparison studies together with upcoming ESA-CCI SIC products, ASI AMSR-E/2 products and MWRI products. We stressed that we used version 3 in this manuscript where appropriate.

*160: Though referenced, it seems simple to actually provide the a and b values in Equation 1 that were used and would be convenient for the reader. It seems these could be potentially added in Table 2?*

This is certainly a good idea, thank you. However, we are speaking about a full set of a and b values because as stated in our manuscript these values are a function of several parameters, one of which is the solar zenith angle varying between the Landsat scenes used. We could include values for several such angles but this would require an additional table. Our paper is already quite long and we believe it is more straightforward to go to that paper and read the values of a and b from the two Figures mentioned in our manuscript. This way a potential user would not be frustrated about the fact that our table – by chance – does not provide those a and b values for the solar zenith angles the user is looking for. Therefore, we kept the information in our paper as is.

*198: I think most readers would be very familiar with the projections/grids, so this is a very minor point: you could provide the EPSG codes (or Proj4 strings) to exactly specify. For example, the NSIDC PS grid (EPSG 3411) is slightly different than another similar WGS84 NSIDC PS grid (EPSG 3413).*

We checked the EPSG codes and provide those we found where appropriate. Note that, e.g. for ASI-SSMI we cannot be sure whether this is EPSG 3411 or 3413 and therefore we don't provide it.

*227: What does "arbitrarily" mean? Were scenes selected randomly? Or did you just pick scenes that looked "good" to use?*

We mean "randomly" and have changed the wording accordingly.

*384-393: You mention snow metamorphism due to melt and melt-refreeze cycles. Another aspect could perhaps be flooded snow and snow-ice formation due to the weight of the snow on the ice causing negative freeboard.*

Yes, certainly this would play a role as well. However, since we refer here to late spring / summer conditions we believe the melt-refreeze and metamorphism plays the more important role here as snow might be damp and/or wet anyways hence mimicking signals emanating from deeper in the snow-pack / the snow-ice system. We added this as a general additional influencing factor, though.

*398: is "ice tongue" the correct term here? I think of an ice tongue as relating to marine-terminating glaciers or ice shelves. You could perhaps use "patch" instead of "tongue", or "floe" or "collection of floes"?*

We changed the wording towards "patch".

*399: the oversampling issue seems quite important here. I assume it is
discussed in the earlier papers referenced, so it isn't necessary to go
into any great detail, but I do think it is worthwhile to mention. One
place would be in the discussion of Table 1 where gridded resolutions
are noted; in that context you could note that the sensor footprint
resolutions are often coarser and thus the effective resolution is
lower (coarser) than the gridded resolution. And then here in line 399,
you could refer back to that to indicate the resolution issue.
Otherwise, as it stands now, this sentence seems to lack context.*

*EDIT: Writing comments as I was reading through it, I hadn't seen
Section 5.1. The addresses the above comment in nice detail. Perhaps
just a reference about the footprint resolution around Table 1 and line
399 noting that it will be discussed in detail in Section 5.1. You
could consider taking the first paragraph of 5.1 and moving it to
Section 2.1, but I can see where that might make that section overly
long.*

Thank you for pointing this out. We actually left the first paragraph
of section 5.1 where it is. We moved the sentence "Note in this context
that we estimate Landsat SIC at the grid resolution of the respective
products, i.e. 12.5 km, 25.0 km or 50.0 km." from lines 401/402 to
right before Table 1 into line 132 and changed it towards: "We estimate
the Landsat SIC at the grid resolution of the respective product (see
Section 2)". We added in Line 401: "We discuss the effect of different
footprint and grid resolutions (see Table 1) in more detail in Section
5.1."

*425: Curious as to the rationale to look at the latter three cases in
the main manuscript rather than the first three? I recognize the
desire/need to limit the length of the main manuscript, which is quite
long. Choosing three is fine, but why those three? Was it just
arbitrary or was there a reason to look at those three over the others?
One thought is that, as noted, the main manuscript is quite long and
these case studies add substantially. I wonder if just selecting one as
a representative example and then relegating the other five to the
supplement might be better? Or perhaps two contrasting cases – e.g.,
freeze-up vs. high-concentration?*

Thank you for this comment closely related to your main overall
concern. Therefore, we'd like to refer to our reply given further up
regarding our choice of cases put into the supplementary material. We
will outline better our choice in the first paragraph of Section 4.

*441-453: I wonder if another reason, in part could be the sampling/sensor resolution issue. At the low resolution of PMW, characteristics can be "smeared" over a larger area. So, high concentration ice could be smeared into lower concentration ice regions, causing a high bias. The fact that this happens as much in the 12.5 km product as in the 25 km maybe argues against this, but it could contribute. I also note in Figure 5 that ASI is particularly low. That suggests there could be some surface and/or atmospheric effect that the 85-90 GHz channels are particularly sensitive to but which may also affect the lower frequency channels?*

Thank you, yes, we agree with you that the resolution plays a role here as well. One issue certainly is the sub-grid scale distribution of ice types. The surface class map does not tell us whether the grey pixels are homogeneous 100% thin ice of, say 10-15 cm thickness (case A) or a mixture of thick, snow covered and young ice of, say 2-5 cm thickness (case B). For case A, equally well-tuned algorithms would provide the same SIC independent of native or grid resolution. For case B, higher resolution SIC products would be impacted by the different surface properties more specifically than low resolution SIC products. However, whether a low resolution SIC based on one BT value integrating over the different surface properties would differ a lot from a mean of a respective number of high resolution SIC values based on several BT values that each represent better the individual surface properties encountered remains to be investigated. Still, we mention this issue in our interpretation of this figure in the revised manuscript.

Note of all ten products ASI SIC has the finest resolution as it is purely based on near-90 GHz data – while SICCI-12km mixes near-90 GHz with 19 GHz data.

We agree that also atmospheric effects could play a role. On the one hand, atmospheric effects would tend to increase SIC – more at higher than at lower frequencies – and could be excluded as a reason to explain the low ASI values. However, on the other hand, the tie points used by the ASI algorithm include a residual, unknown weather influence (on purpose actually) – especially the open water tie point. If the atmosphere is drier than what is implicitly included in the open water tie point, then the SIC retrieved by the ASI algorithm could actually be lower than the true SIC. We note that this is an issue potentially impacting all algorithms but is possibly most pronounced for algorithms such as the ASI algorithm using fixed tie point values (see also Andersen et al., 2006, Remote Sensing of Environment, 104(4), 374-392). We add this information in the revised manuscript.

Finally, we note here in our reply the particularly low SIC values for ASI cannot be explained with the presence of young / thin sea ice because at higher frequencies the sensitivity of the SIC to thin ice is smaller than at lower frequencies.

*Technical Corrections (by line number):*

*58: suggest omitting "for sure" or substituting something like "clearly" or "definitely"*

Changed as suggested.

*79: "1980s" not "1980ties"*

Changed as suggested.

*232: suggest "Thus," instead or "With that"*

Changed as suggested.

*397: "…of, for example, a…"*

Re-ordered as suggested.

*430: "We have only a few" instead of "We have got only few"*

Changed as suggested.

*439: "Contrary to expectations" instead of "Unlike expected"*

Changed as suggested.

*525: Table 10, "behind the / values"? Do you mean to the right of the "/"? That would be better terminology "to the right" (or left) not "behind". Similarly in Lines 548, 564, and 623.*

Thank you. We used the improved terminology suggested by the reviewer.

*780: suggest "derive" or "make" instead of "come up with"*

Changed as suggested.